# Cystatin F (*Cst7*) drives sex-dependent changes in microglia in an amyloid-driven model of Alzheimer's disease

**Michael JD Daniels[1]\*, Lucas Lefevre[1], Stefan Szymkowiak[1], Alice Drake[1], Laura McCulloch[1,2], Makis Tzioras[1], Jack Barrington[1], Owen R Dando[1,3], Xin He[1,3], Mehreen Mohammad[1], Hiroki Sasaguri[4,5], Takashi Saito[5,6,7], Takaomi C Saido[5], Tara L Spires-Jones[1], Barry W McColl[1]\***

[1]UK Dementia Research Institute at The University of Edinburgh, Edinburgh, United Kingdom; [2]Centre for Inflammation Research, Institute for Regeneration and Repair, The University of Edinburgh, Edinburgh, United Kingdom; [3]Simons Initiative for the Developing Brain, University of Edinburgh, Edinburgh, United Kingdom; [4]Department of Neurology and Neurological Science, Graduate School of Medicine, Tokyo Medical and Dental University, Tokyo, Japan; [5]Laboratory for Proteolytic Neuroscience, RIKEN Brain Science Institute, Wako, Japan; [6]Department of Neuroscience and Pathobiology, Research Institute of Environmental Medicine, Nagoya University, Nagoya, Japan; [7]Department of Neurocognitive Science, Institute of Brain Science, Nagoya City University Graduate School of Medical Sciences, Nagoya, Japan

**\*For correspondence:**
mdaniels@ed.ac.uk (MJDD);
barry.mccoll@ed.ac.uk (BWMcC)

**Abstract** Microglial endolysosomal (dys)function is strongly implicated in neurodegenerative disease. Transcriptomic studies show that a microglial state characterised by a set of genes involved in endolysosomal function is induced in both mouse Alzheimer's disease (AD) models and human AD brain, and that the emergence of this state is emphasised in females. *Cst7* (encoding cystatin F) is among the most highly upregulated genes in these microglia. However, despite such striking and robust upregulation, the function of *Cst7* in neurodegenerative disease is not understood. Here, we crossed *Cst7*[-/-] mice with the *App*[NL-G-F] mouse to test the role of *Cst7* in a model of amyloid-driven AD. Surprisingly, we found that *Cst7* plays a sexually dimorphic role regulating microglia in this model. In females, *Cst7*[-/-]*App*[NL-G-F] microglia had greater endolysosomal gene expression, lysosomal burden, and amyloid beta (Aβ) burden *in vivo* and were more phagocytic *in vitro*. However, in males, *Cst7*[-/-]*App*[NL-G-F] microglia were less inflammatory and had a reduction in lysosomal burden but had no change in Aβ burden. Overall, our study reveals functional roles for one of the most commonly upregulated genes in microglia across disease models, and the sex-specific profiles of *Cst7*[-/-]-altered microglial disease phenotypes. More broadly, the findings raise important implications for AD including crucial questions on sexual dimorphism in neurodegenerative disease and the interplay between endolysosomal and inflammatory pathways in AD pathology.

## Editor's evaluation

This study presents a valuable finding on the function of the gene Cst7 in sex-divergent pathological changes in microglia in a mouse model of amyloid-driven Alzheimer's disease. The evidence supporting the claims of the authors is solid, although the study would be further strengthened by validation of some of the key differentially expressed genes identified in RNA-sequencing experiments. Overall, this study offers new insight into the functional role of CST7 that is upregulated in a

subset of disease-associated microglia in AD models and the human brain that will be of interest to neuroimmunologists and neuroscientists working on microglia in health and disease.

## Introduction

Alzheimer's disease (AD) is a chronic neurodegenerative disease characterised pathologically by build-up of protein aggregates such as amyloid beta (Aβ) and hyperphosphorylated tau. AD is also characterised by reactive glial cells, including the emergence of altered microglia. Microglia are the predominant immune cells of the brain parenchyma and play numerous roles in development and homeostasis (*Colonna and Butovsky, 2017*). In recent years, microglia have become heavily implicated in the pathogenesis of AD, in part due to accumulating genetic and epigenetic evidence that single nucleotide polymorphisms leading to altered risk for late-onset AD (LOAD) are enriched in microglial-expressed genes associated with endolysosomal function and lipid handling (*Podleśny-Drabiniok et al., 2020*). This implies that microglia may play a causal role in the disease and supports the theory that microglia may be targeted to treat AD. In addition to genetics, sex is a major risk factor for AD, with women up to twice as likely to be diagnosed than men (*Podcasy and Epperson, 2016*). This sex bias is not entirely due to increased longevity in females, as increased risk remains after adjusting for lifespan, and may be due to interaction effects between sex/gender and genetics (*Gamache et al., 2020*). The mechanisms for this, and possible interplay between microglia and sex are poorly understood.

Recently, an altered state of microglia has been identified in AD models that arises dependent on AD risk gene *TREM2*. These cells, termed disease-associated microglia (DAM)/microglial neurodegenerative phenotype (MGnD)/activated response microglia (ARM) (*Keren-Shaul et al., 2017*; *Krasemann et al., 2017*; *Sala Frigerio et al., 2019*), are characterised by downregulation of microglial 'homeostatic' genes *Tmem119* and *P2ry12* and upregulation of genes such as *Apoe*, *Spp1*, *Itgax*, *Clec7a,* and *Cst7*. Notably, the emergence of this subset may be accelerated in female mice compared to male (*Sala Frigerio et al., 2019*). DAM/MGnD/ARM are believed to be highly phagocytic cells due to the enrichment of phagocytic and lipid-handling pathways in their gene sets (*Deczkowska et al., 2018*). However, the precise functions of many of the genes that make up DAM/MGnD/ARM are poorly understood.

*Cst7*, coding for the protein cystatin F (CF), is amongst the most robustly upregulated genes in the DAM/MGnD/ARM signature. *Cst7*/CF expression appears to be driven by amyloid plaques as upregulation at both the RNA and protein level is spatially localised to plaques (*Ofengeim et al., 2017*; *Chen et al., 2020*). CF is atypical in that it is localised to the endolysosomal compartment where it acts as an endogenous inhibitor of cysteine proteases such as cathepsin L and C (*Hamilton et al., 2008*). However, little is understood about the mechanistic role of CF in the context of central nervous system (CNS) disease. Some studies have suggested CF may play a protective role in a demyelination model by inhibiting cathepsin C (*Liang et al., 2016*) or negatively regulating microglial phagocytosis (*Kang et al., 2018*; *Popescu et al., 2023*) but the possible disease-modifying role of *Cst7*/CF in amyloid-driven neurodegenerative disease is unknown. Additionally, although there is some evidence that *Cst7* itself is upregulated to a greater extent in female vs. male mice (*Kang et al., 2018*; *Sala Frigerio et al., 2019*; *Guillot-Sestier et al., 2021*), the sex-specific role of *Cst7* in disease has not been investigated.

Here, we set out to investigate what, if any, disease-modifying role *Cst7*/CF may play in an amyloid-driven AD mouse model in both females and males. We first compiled data from published datasets to show that *Cst7* is indeed robustly and consistently upregulated in microglia across numerous amyloid-driven models of AD. We then deleted *Cst7* in male and female *App^{NL-G-F}* knock-in mice which showed that *Cst7* drives sex-dependent changes in microglia at transcript, protein, and functional level. *Cst7* deletion in male mice led to downregulation of inflammatory transcripts and reduced lysosomal burden but had no effect on microglial Aβ burden *in vivo* or plaque deposition. However, in females, *Cst7* deletion led to increased expression of endolysosomal genes, increased lysosomal burden, and increased microglial Aβ burden *in vivo*. We further investigated this *in vitro* and showed that this is due to increased phagocytosis rather than impaired degradation. These data bring important insight to the functional role of *Cst7*, a key hallmark gene within the DAM/MGnD/ARM disease microglia

signature. They also highlight that pathology-influencing effects are sex-dependent, underlining the interaction between sex and microglial function relevant to neurodegenerative disease.

## Results

### *Cst7*/CF is upregulated in female and male microglia in murine models of AD

*Cst7* is often detected in RNASeq or scRNASeq databases as significantly upregulated in disease models. However, an integrated analysis of *Cst7* expression in disease models stratified by sex is lacking. Therefore, we searched publicly available datasets for *Cst7* expression in microglia in mouse models of AD. We integrated data from six studies in which microglial *Cst7* expression was profiled (*Figure 1A*). Expression data were restricted to microglia by use of techniques such as fluorescence-activated cell sorting (FACS) (*Orre et al., 2014*; *Wang et al., 2015*; *Srinivasan et al., 2020*), immuno-magnetic separation (*Guillot-Sestier et al., 2021*), RiboTag translational profiling (*Kang et al., 2018*), or single-cell RNA sequencing (*Sala Frigerio et al., 2019*). We were able to collect data on *Cst7* expression in males *vs.* females in the APP/PS1 (*Guillot-Sestier et al., 2021*) and *App*^NL-G-F model (*Sala Frigerio et al., 2019*). In all models, *Cst7* was dramatically upregulated in microglia in amyloid-driven disease and in models where we were able to compare sexes, *Cst7* appeared marginally more highly expressed in female disease brains than in male. This is consistent with literature suggesting the DAM/ARM/MGnD microglial subset is accelerated in female mice (*Sala Frigerio et al., 2019*). We further investigated sexual dimorphism in microglial states by sub-setting ARMs from Sala-Frigerio et al. and discovered, alongside sex-chromosome genes, endolysosomal genes such as *Spp1* and *Gpnmb* amongst *Cst7* as marginally more expressed in female ARMs *vs.* male ARMs (*Figure 1—figure supplement 1A*). To investigate cell-specific expression of *Cst7*, we mined sequencing databases and found *Cst7* is expressed almost exclusively in disease-derived microglial clusters with some minimal expression in homeostatic microglia and negligible expression in other CNS or immune cell subtypes (*Zhang et al., 2014*; *Van Hove et al., 2019*; *Zeng et al., 2023*; *Figure 1—figure supplement 1B–D*). Finally, we investigated the expression of *Cst7* in intact tissue by conducting *in situ* hybridisation for *Cst7* with markers for IBA1 and Aβ plaques for localisation. We found that *Cst7* was dramatically upregulated in disease where expression decorated plaques in *App*^NL-G-F brains (*Figure 1B*). Furthermore, expression was significantly enriched around plaques (*Figure 1C and D*) and almost exclusively localised to IBA1+ cells surrounding plaques (*Figure 1E and F*). Together, these data validate *Cst7* as a key marker gene of disease-induced reactive microglia, notably of the DAM/MGnD/ARM state, across cerebral amyloidosis models in both sexes, and suggest that its disease-induced upregulation is related temporally and spatially to Aβ plaque pathology.

### *Cst7* deletion leads to sex-dependent transcriptomic changes in microglia in the *App*^NL-G-F model of amyloid-driven AD

As *Cst7* is so robustly upregulated in microglia in amyloid-driven disease and some studies suggest this to be greater in female mice, we next aimed to investigate whether *Cst7*/CF plays a role in regulating microglia in disease and whether this regulation could be sex-dependent. Therefore, we crossed *Cst7*^-/- mice with the *App*^NL-G-F model of amyloid-driven AD (*Saito et al., 2014*) to generate background-matched non-disease (*App*^Wt/Wt*Cst7*^+/+), disease (*App*^NL-G-F*Cst7*^+/+) and disease knockout (*App*^NL-G-F*Cst7*^-/-) mice. As *Cst7* expression is limited to the myeloid compartment in the brain (*Zhang et al., 2014*), this study primarily tests the role of microglial/border-associated macrophages in amyloid-driven disease. We first investigated the role of *Cst7*/CF in regulating the microglial response in disease by unbiased analysis of the microglial transcriptome. Mice were aged to 12 months to drive Aβ plaque burden and a DAM/ARM microglial profile that includes marked upregulation of *Cst7* before microglia were isolated from hemi-brains by FACS (*Figure 2A*). Microglia were defined by FSC/SSC profile, singlet, live, CD11b+, Ly6C-, CD45+ (*Figure 2—figure supplement 1A*) and sorted along with phenotypic markers P2Y12, MHCII, and CD11c (*Figure 2—figure supplement 1*) before RNA purification, sequencing, and analysis using DESeq2 (*Love et al., 2014*).

We first assessed disease-induced microglial changes by comparing *App*^Wt/Wt*Cst7*^+/+ and *App*^NL-G-F*Cst7*^+/+ microglia in male and female mice. *App*^NL-G-F knock-in led to numerous changes in gene transcription with 2022 genes upregulated and 1849 downregulated in males and 2053 upregulated

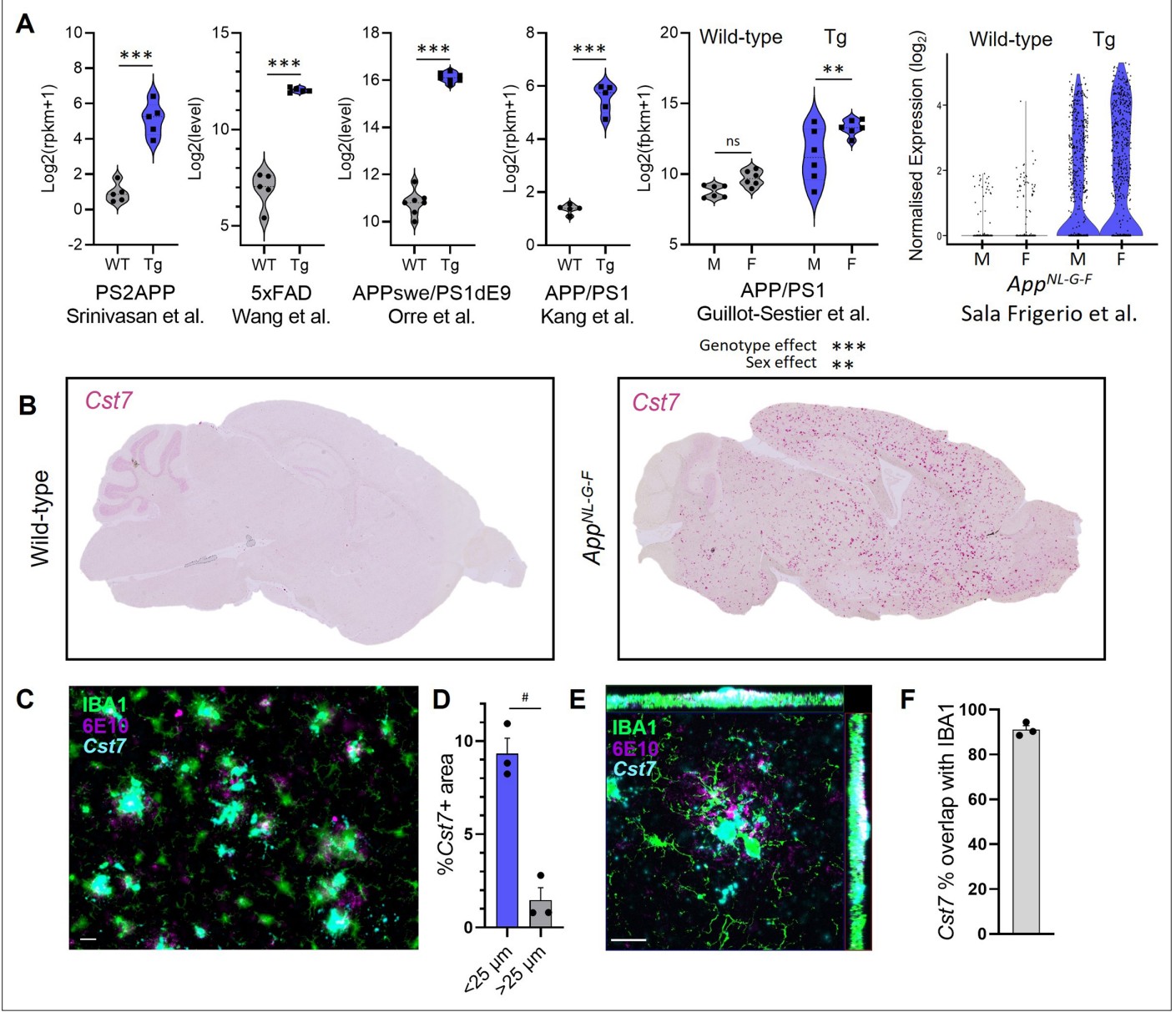

**Figure 1.** *Cst7*/cystatin F is upregulated in microglia in murine models of Alzheimer's disease (AD). (**A**) Analysis of publicly available microglial RNASeq databases for *Cst7* expression in amyloid-driven models. Background-matched control (grey) or relevant disease model (blue). (**B**) Brightfield slide-scanned images of wild-type (left) and *App^NL-G-F* (right) sagittal brain sections stained by *in situ* hybridisation for *Cst7* (red). (**C**) Example image from *Cst7* fluorescence *in situ* hybridisation (FISH) experiment stained with IBA1 (green), 6E10 (magenta), and *Cst7* (cyan). (**D**) Quantification of (**C**) showing % *Cst7*+ area in areas within 25 µm (blue) of 6E10+ plaques or outside (grey). (**E**) Hi-mag confocal microscopy maximum intensity projection showing co-localisation of *Cst7* with IBA1 around plaques. (**F**) Quantification of (**E**) showing % *Cst7* overlap with IBA1. n=3–7. **p<0.01, ***p<0.001 by unpaired t-test or two-way ANOVA with Tukey's multiple comparisons post hoc test. #p<0.05 by paired t-test. Scale bars are 20 µm.

The online version of this article includes the following source data and figure supplement(s) for figure 1:

**Source data 1.** Source data associated with *Figure 1*.

**Figure supplement 1.** Investigating expression of *Cst7*.

and 1857 downregulated in females (*Figure 2B*). Differentially regulated genes were mostly shared between males and females but we did observe uniquely regulated genes between sexes (*Figure 2—figure supplement 2B*). As expected, upregulated genes included DAM/ARM genes such as *Clec7a*, *Spp1*, *Itgax*, and *Csf1*, whilst downregulated genes included microglial 'homeostatic' genes *P2ry12*, *Tmem119*, and *Gpr34* in both sexes (*Figure 2C and D*). *Cst7* was among the top 10 most highly

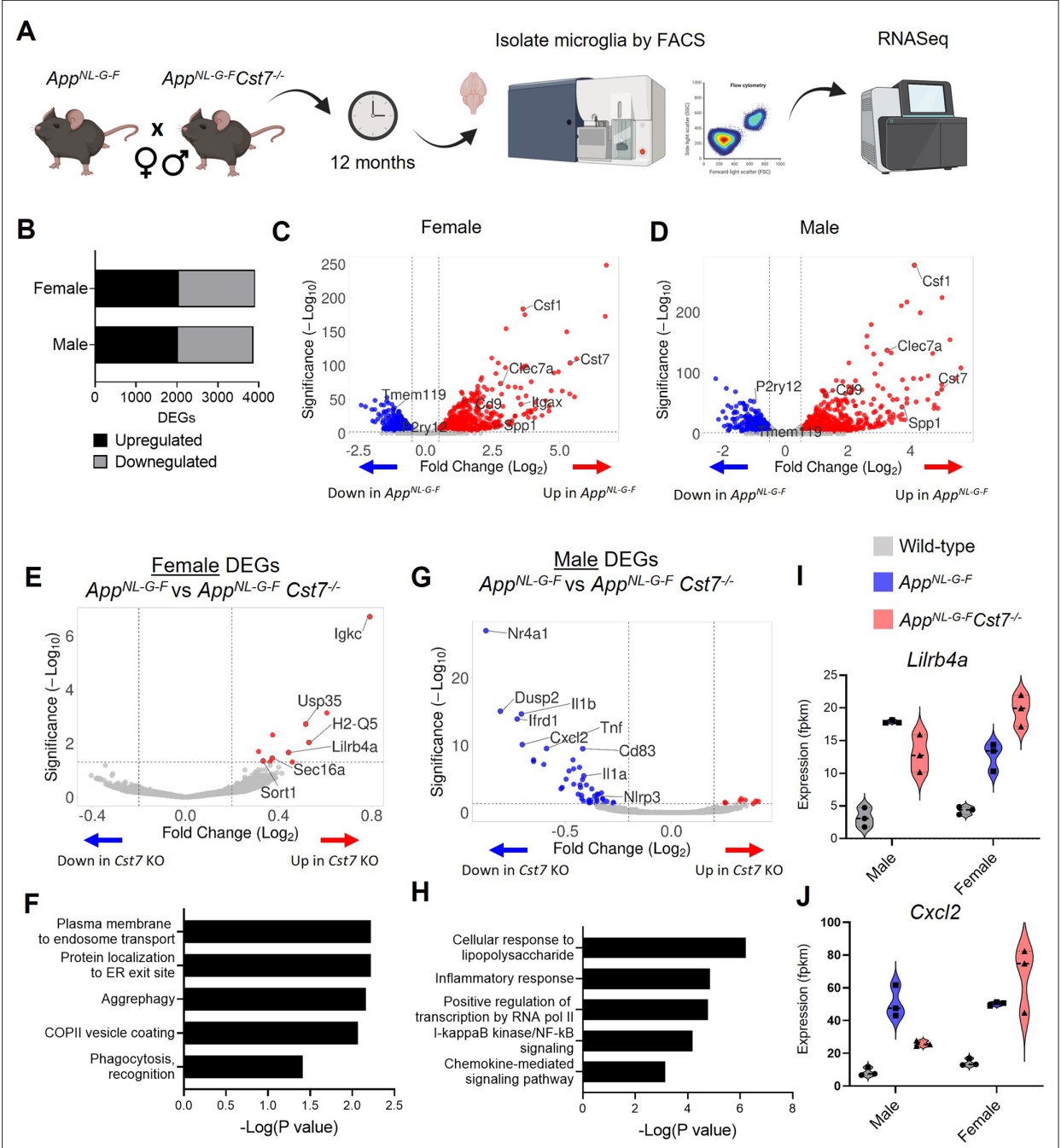

**Figure 2.** *Cst7* deletion leads to sex-dependent transcriptomic changes in microglia in the *App*<sup>NL-G-F</sup> model of amyloid-driven Alzheimer's disease (AD). (**A**) Study design schematic. (**B**) Differentially expressed genes (DEGs) between wild-type and *App*<sup>NL-G-F</sup> microglia in male and female mice. (**C**) Volcano plot of RNASeq from wild-type *vs. App*<sup>NL-G-F</sup> female microglia. Points in red are significantly upregulated in disease. Points in blue are significantly downregulated. Selected genes are annotated. (**D**) Volcano plot of RNASeq from wild-type *vs. App*<sup>NL-G-F</sup>*Cst7*<sup>+/+</sup> male microglia. (**E**) Volcano plot of RNASeq from female *App*<sup>NL-G-F</sup>*Cst7*<sup>+/+</sup> *vs. App*<sup>NL-G-F</sup>*Cst7*<sup>-/-</sup> microglia. Points in red are significantly upregulated in *App*<sup>NL-G-F</sup>*Cst7*<sup>-/-</sup>. Points in blue are significantly downregulated. Selected genes are annotated. (**F**) Selected GO:BP (Gene Ontology:biological process) terms that are significantly enriched in the upregulated genes from (**E**) and corresponding significance p-value. (**G**) Volcano plot of RNASeq from male *App*<sup>NL-G-F</sup>*Cst7*<sup>+/+</sup> *vs. App*<sup>NL-G-F</sup>*Cst7*<sup>-/-</sup> microglia. (**H**) Selected GO:BP terms that are significantly enriched in the upregulated genes from (**G**) and corresponding significance p-value. (I and J) Example gene expression (fpkm) from RNASeq of microglia isolated from male and female wild-type, *App*<sup>NL-G-F</sup>*Cst7*<sup>+/+</sup>, and *App*<sup>NL-G-F</sup>*Cst7*<sup>-/-</sup> mice. Selected genes are *Lilrb4* (**I**) and *Cxcl2* (**J**).

The online version of this article includes the following source data and figure supplement(s) for figure 2:

**Source data 1.** Source data associated with *Figure 2*.

*Figure 2 continued on next page*

*Figure 2 continued*

**Source data 2.** Differentially expressed genes (DEGs) between male and female App^NL-G-F and *App^NL-G-F*Cst7^-/- mice (selected genes in bold).

**Figure supplement 1.** Gating strategy for microglia fluorescence-activated cell sorting (FACS).

**Figure supplement 2.** Gene expression from RNASeq data.

**Figure supplement 3.** Gene expression from RNASeq data compared to qPCR from naïve (non-diseased) mice.

upregulated genes in males and females. Interestingly, although there were relatively few differentially expressed genes (DEGs) between male and female mice in the *App^NL-G-F*Cst7^+/+ group (*Figure 2—source data 2*), these genes did include DAM/ARM genes such as *Gpnmb*, *Spp1,* and *Ctse*. This is consistent with the observation that the DAM/ARM profile is accelerated in females compared to males (*Figure 1A*; *Sala Frigerio et al., 2019*).

Having confirmed the upregulation of the *Cst7* and the DAM/ARM profile in this model, we next tested the role of *Cst7* in disease by comparing differential expression between the *App^NL-G-F*Cst7^+/+ and *App^NL-G-F*Cst7^-/- groups in both sexes. Importantly, *Cst7* itself was the most DEG in both sexes, confirming successful deletion (*Figure 2—figure supplement 2A*). *Cst7* deletion led to significant changes in microglial transcriptome. In females, *Cst7* knockout led to upregulation of 15 genes and downregulation of 2 genes (including *Cst7*) (*Figure 2E*). Despite the modest number of DEGs, we observed that many of the upregulated genes appeared to be involved in endolysosomal-related processes such as protein trafficking to organelles (*Sort1*, *Sec16a*) and antigen presentation (*Lilrb4a*, *Igkc*). Enrichment analyses revealed that female *App^NL-G-F*Cst7^-/- microglia were significantly enriched in GO:BP (Gene Ontology: biological process) terms such as 'plasma membrane to endosome transport', 'aggrephagy', and 'phagocytosis' against their *App^NL-G-F*Cst7^+/+ counterparts (*Figure 2F*).

Next, we tested whether *Cst7* knockout led to the same effect in males as in females. Surprisingly, we observed that male *App^NL-G-F*Cst7^-/- microglia exhibited markedly different transcriptome compared to *App^NL-G-F* with upregulation of 8 genes and downregulation of 51 genes (including *Cst7*) (*Figure 2G*). In contrast to females, downregulated genes in males were dominated by classical proinflammatory mediators such as *Il1b*, *Il1a*, *Tnf*, and *Cxcl2* and enrichment analyses revealed downregulation of GO:BP terms such as 'cellular response to lipopolysaccharide', inflammatory response', and 'NF-kappaB signaling' (*Figure 2H*). There was very little overlap between *Cst7*-regulated genes in males and females with only two genes shared including *Cst7* (*Figure 2—figure supplement 2C*). Notably, many disease-induced genes followed an intriguing pattern of being suppressed in *App^NL-G-F*Cst7^-/- vs. *App^NL-G-F*Cst7^+/+ in males but potentiated in *App^NL-G-F*Cst7^-/- vs. *App^NL-G-F*Cst7^+/+ in females. Examples of this are immunoglobulin receptor gene *Lilrb4a* and inflammatory gene *Cxcl2* (*Figure 2I and J*). Interestingly, *Cst7* also appeared to drive a sex-genotype interaction effect. We determined male *vs.* female DEGs in *App^NL-G-F*Cst7^+/+ and their *App^NL-G-F*Cst7^-/- counterparts. In *App^NL-G-F*Cst7^+/+ microglia, there were 33 DEGs between male vs. female approximately evenly split between transcripts enriched in males *vs.* those enriched in females, whereas in *App^NL-G-F*Cst7^-/- we observed 240 DEGs dominated by those enriched in females (*Figure 2—figure supplement 2D*). These transcripts included inflammatory genes such as *Il1b*, *Tnf,* and *Cxcl2*.

Finally, we sought to validate our findings through complimentary methods and confirm the gene changes we observe are microglia-expressed. We used fluorescence *in situ* hybridisation (FISH) coupled with immunostaining to measure expression of *Lilrb4a* in non-processed brain tissue. We confirmed expression of *Lilrb4a* in microglia specifically around plaques and validated our RNASeq data showing an identical expression pattern to that observed previously with increased expression of microglial *Lilrb4a* in *App^NL-G-F*Cst7^-/- vs. *App^NL-G-F*Cst7^+/+ in females but not in males (*Figure 2—figure supplement 3E and F*). We also assessed expression of key hit genes *Lilrb4a* and *Il1b* by performing qPCR in microglia isolated from non-diseased wild-type vs *Cst7*^-/- mice to confirm disease-specific regulation. Indeed, we did not see the same pattern as described in RNASeq with upregulation in disease and further upregulation in *Cst7*^-/- in females (*Lilrb4a*) or downregulation in *Cst7*^-/- males (*Il1b*) (*Figure 2—figure supplement 3A*). In fact, there were no significant differences between groups (*Figure 2—figure supplement 3B*).

Together, these data show that *Cst7* drives sex-dependent effects on microglial transcriptome in an amyloid-driven mouse model of AD. *Cst7* knockout led to a disease-dependent upregulation of a small set of endolysosomal-enriched genes in female microglia but a striking downregulation of

inflammatory genes in male microglia. The number of DEGs between $App^{NL-G-F}$ males and females was markedly greater in the $Cst7^{-/-}$ state, implying a sex-genotype interaction effect, that is, the impact of $Cst7$ deficiency on microglial phenotype in $App^{NL-G-F}$-driven disease differs quantitatively and qualitatively (e.g. in endolysosomal and inflammatory pathways) according to sex.

## $Cst7$ deletion leads to region and sex-dependent changes in lysosomes in $App^{NL-G-F}$ mice

Following our investigation into the $Cst7$-dependent effects on microglia at the transcriptome level, we next sought to study changes at the protein level with spatial resolution. This allowed us to investigate brain region-dependent effects. We took sagittal brain sections from 12-month male and female $App^{Wt/Wt}Cst7^{+/+}$, $App^{NL-G-F}Cst7^{+/+}$, and $App^{NL-G-F}Cst7^{-/-}$ mice and immunostained for IBA1 to visualise microglia/macrophages, anti-Aβ clone 6E10 to visualise Aβ, and, as both previous literature and our RNASeq data in *Figure 2* implicated lysosomes in $Cst7$ function, LAMP2 to visualise lysosomes. We assessed three brain regions: cortex, whole hippocampus, and the dorsal subiculum (*Figure 3—figure supplement 1A*), to investigate the effects of disease and $Cst7$ knockout on microglial burden and lysosomal activation. As expected, knock-in of the $App^{NL-G-F}$ construct led to marked build-up of 6E10+ Aβ plaques, an increase in IBA1+ microglial burden around these plaques, and upregulation of LAMP2 in microglia predominantly in plaque-associated microglia in both males and females (*Figure 3A*). LAMP2 predominantly co-localised with IBA1, suggesting that expression is predominant in mononuclear phagocytes rather than neuronal cells or macroglia. Next, we compared $App^{NL-G-F}Cst7^{+/+}$ vs. $App^{NL-G-F}Cst7^{-/-}$ male and female mice. We observed a trend towards increased IBA1+ microglial burden in $App^{NL-G-F}Cst7^{-/-}$ mice in all brain regions, statistically significant in the subiculum (*Figure 3B–D*), more evidently in male mice. Importantly, this was not only due to an increase in Aβ plaque load (which correlated with microglial burden) as images were quantified in areas of even plaque coverage (*Figure 3—figure supplement 1B–D*).

To investigate the effect of $Cst7$ deletion on lysosomal activity, we quantified LAMP2 immunostaining and normalised to IBA1 as these markers are highly correlated (*Figure 3—figure supplement 1E*) and we wanted to remove the confounding factor of a greater IBA1 burden in $Cst7$ knockout mice (*Figure 3B–D*). Strikingly, we found in the subiculum that microglial-normalised LAMP2 staining followed the same pattern as many of the genes identified by microglia RNASeq with reduced expression in $App^{NL-G-F}Cst7^{-/-}$ vs. $App^{NL-G-F}Cst7^{+/+}$ males but increased expression in $App^{NL-G-F}Cst7^{-/-}$ vs. $App^{NL-G-F}Cst7^{+/+}$ females (*Figure 3E*, quantification in *Figure 3H*). A similar pattern was observed in the cortex and hippocampus but without a significant change in females (*Figure 3F and G*). We hypothesised that the reason for this region-dependent effect may be due to differences in plaque structure. Aβ plaques typically comprise a β-sheet-containing dense core and a more diffuse outer ring (*DeTure and Dickson, 2019*). We utilised Congo Red derivative methoxy-X04 (MeX04), which binds selectively to fibrillar β-sheet-rich structures such as the dense cores of amyloid plaques (*DeTure and Dickson, 2019*), compared to anti-Aβ antibody 6E10, which will bind both dense cores and diffuse outers to study plaque composition in different brain regions of $App^{NL-G-F}$ mice (*Figure 3—figure supplement 1A*). MeX04 was injected intraperitoneally 2.5 hr before mice were sacrificed and brains taken for histology. We found that Aβ plaques had a far greater MeX04:6E10 ratio in the subiculum compared to other brain regions (*Figure 3I and J*, quantified in *Figure 3K*), most likely caused by earlier Aβ accumulation in this region (*Rönnbäck et al., 2011*; *Gail Canter et al., 2019*) that may be dependent on a subpopulation of microglia (*Spangenberg et al., 2019*). This implies that the sex-dependent effects of $Cst7$ deletion on microglial lysosomes may be driven by plaque composition/structure, with regions dominated by dense core plaques (indicating further 'disease progression') showing greater effect of $Cst7$ deletion.

## $Cst7$ regulates microglial Aβ burden in female $App^{NL-G-F}$ mice

After discovering that $Cst7$ deletion leads to sex-dependent changes in microglia at both the transcriptome and protein level indicative of an altered endolysosomal system, we next further investigated potential functional changes in these altered microglia. A major function of microglia in the context of AD is uptake of Aβ into lysosomes (*Grubman et al., 2021*). To study this, we utilised MeX04 injection as described previously to label Aβ-containing microglia in the brain. First, we validated that intraperitoneal injection of MeX04 2.5 hr before microglial isolation by FACS can detect Aβ-containing

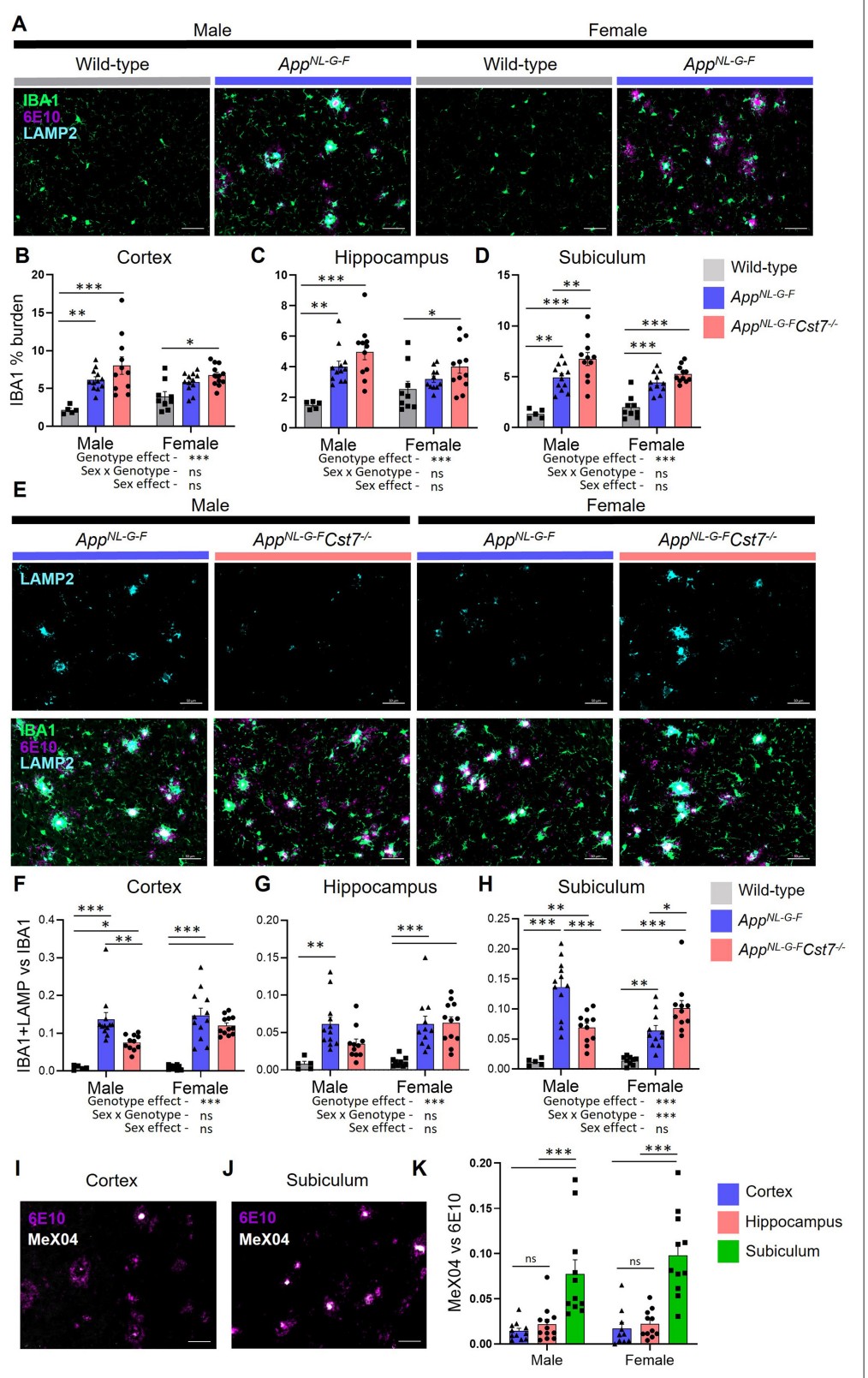

**Figure 3.** *Cst7* deletion leads to region and sex-dependent changes in lysosomes in *App*^*NL-G-F*^ mice. (**A**) Example images from male and female wild-type and *App*^*NL-G-F*^ cortex stained with IBA1 (green), 6E10 (magenta), and LAMP2 (cyan). (**B–D**) IBA1 % coverage from images taken of wild-type, *App*^*NL-G-F*^*Cst7*^*+/+*^, and *App*^*NL-G-F*^*Cst7*^*-/-*^ brains in cortex (**B**), hippocampus (**C**), and dorsal subiculum (**D**). (**E**) Example images from *App*^*NL-G-F*^*Cst7*^*+/+*^ and *App*^*NL-G-*^

*Figure 3 continued on next page*

*Figure 3 continued*

[F]*Cst7*[-/-] subicula in male and female mice. Top panel shows LAMP2 (cyan) and bottom panel shows IBA1 (green), 6E10 (magenta), and LAMP2 (cyan) merge. Scale bars are 50 μm. (**F–H**) Ratio of IBA1/LAMP2 double positive staining *vs.* IBA1 total staining in the cortex (**F**), hippocampus (**G**), and dorsal subiculum (**H**) of wild-type, *App*[NL-G-F]*Cst7*[+/+], and *App*[NL-G-F]*Cst7*[-/-] mice. (I and J) Example images from the cortex (**I**) and subiculum (**J**) of an *App*[NL-G-F] mouse stained with 6E10 (magenta) and MeX04 (white). (**K**) Ratio of MeX04 total staining *vs.* 6E10 total staining in cortex (blue), hippocampus (red), and subiculum (green) of male and female *App*[NL-G-F] mice. Scale bars are 50 μm. Data points are average from 2 fields of view/mouse (1 for subiculum) and bars are plotted as mean + SEM. n=5–12. *p<0.05, **p<0.01, ***p<0.001 by two-way ANOVA with Tukey's multiple comparisons post hoc test.

The online version of this article includes the following source data and figure supplement(s) for figure 3:

**Source data 1.** Source data associated with *Figure 3*.

**Figure supplement 1.** Immunohistochemistry supplementary information.

microglia. Wild-type and *App*[NL-G-F] mice were aged to 12 months and injected with either MeX04 or vehicle before microglial isolation by FACS as previously but with additional detection of MeX04 in the DAPI channel. As expected, wild-type+MeX04 and *App*[NL-G-F]+vehicle mice displayed a background MeX04 signal that was shifted only in the *App*[NL-G-F]+MeX04 group (*Figure 4—figure supplement 1*).

After validating our approach, we next investigated the role of *Cst7* on microglial Aβ burden (*Figure 4A*). Consistent with our RNASeq and histology data suggesting increased lysosomal capacity specifically in female *App*[NL-G-F] mice following deletion of *Cst7*, we observed a greater Aβ burden in female *App*[NL-G-F]*Cst7*[-/-] microglia compared to *App*[NL-G-F]*Cst7*[+/+] counterparts measured by both % Aβ-positive microglia (*Figure 4B*) and mean fluorescence intensity of MeX04 on microglia (*Figure 4C*). Interestingly, despite a reduced expression of lysosomal membrane marker LAMP2 in male *App*[NL-G-F]*Cst7*[-/-] mice compared to *App*[NL-G-F]*Cst7*[+/+] counterparts (*Figure 3C and D*), we observed no difference in microglial Aβ burden (*Figure 4B*). This may be due to the counteracting effect of a reduction of inflammatory genes such as *Il1b* and *Nlrp3* (*Figure 2G*) which have been shown to impair phagocytosis (*Heneka et al., 2013*; *Tejera et al., 2019*).

The benefit of using a FACS-based approach to investigating microglial Aβ burden is the opportunity to include surface phenotypic markers to assess microglial reactivity. We assessed expression of CD45, a broad but sensitive marker of activation intensity within microglia; P2Y12, a microglial homeostatic marker; CD11c, a marker shown to be upregulated in disease/damage-associated microglia (*Keren-Shaul et al., 2017*) and MHCII, a marker of antigen presentation, a process which is also dependent on peptide processing through the endolysosomal system (*Figure 2—figure supplement 1*). *App*[NL-G-F] microglia had increased expression of CD45, CD11c, and MHCII compared to wild-type counterparts (*Figure 4D–F*). We did not detect a reduction in P2Y12 expression (*Figure 4G*). *Cst7* deletion did not lead to any significant differences in detection of these markers, although we did note a trend towards increased MHCII+ microglia only in female mice, perhaps consistent with elevated endolysosomal activity and our RNASeq data suggesting an increase in antigen presentation pathway genes specifically in this group.

## *Cst7* negatively regulates phagocytosis in microglia from female *App*[NL-G-F] mice

Our current data suggest that *Cst7* plays a sex-dependent regulatory role on microglia leading to increased endolysosomal gene pathways, region-dependent lysosomal activity, and Aβ burden in female *App*[NL-G-F]*Cst7*[-/-] microglia. However, *in vivo* it is challenging to ascertain whether the increase in microglial Aβ is due to an increased uptake or to an impaired degradation. To test this, we conducted *in vitro* assays on *Cst7*[+/+] and *Cst7*[-/-] microglia to examine phagolysosomal and related functions. Initially, we examined microglia from young adult *Cst7*[+/+] and *Cst7*[-/-] background-matched mice to understand any baseline differences in the absence of a disease exposure. Unlike constitutively highly expressed microglial genes such as *Trem2* where knockout leads to dramatic alterations in *in vitro* phenotype (*Takahashi et al., 2005*; *Hsieh et al., 2009*; *Kleinberger et al., 2014*; *Yeh et al., 2016*), *Cst7* is expressed at very low levels basally (*Sala Frigerio et al., 2019*; *Figure 1A*). Therefore, we reasoned that *Cst7* deletion would have limited impact on microglial phenotype at baseline where *Cst7* is not highly expressed. Comparing *Cst7*[+/+] and *Cst7*[-/-] microglia, we observed no significant difference in uptake of Aβ oligomers, human AD synaptoneurosomes, *Staphylococcus aureus*

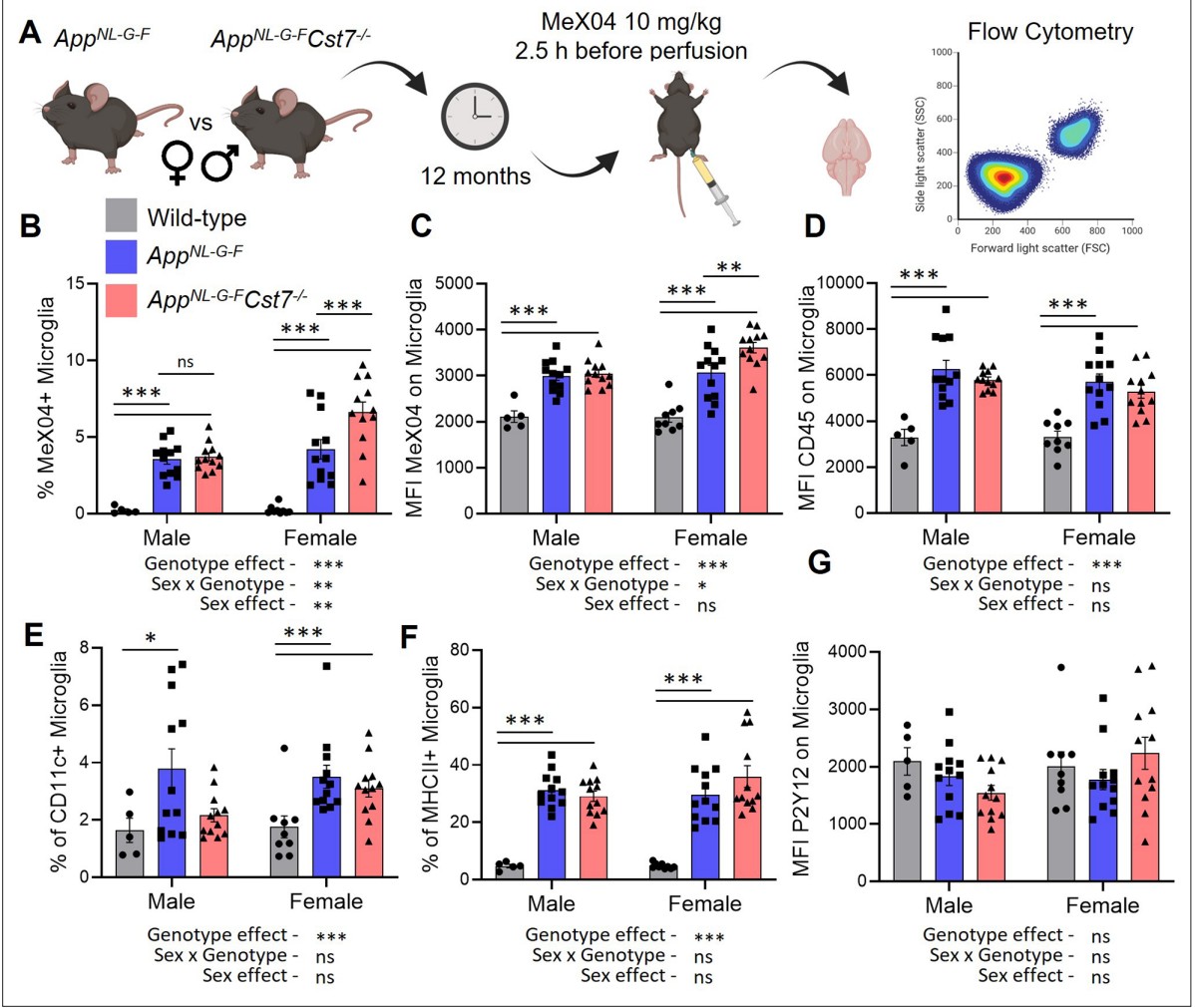

**Figure 4.** *Cst7* regulates microglial Aβ burden in female *App*^*NL-G-F*^ mice. (**A**) Study design schematic. (**B**) % MeX04+ microglia of total microglia in male and female wild-type (grey), *App*^*NL-G-F*^*Cst7*^*+/+*^ (blue), and *App*^*NL-G-F*^*Cst7*^*-/-*^ (red) brains. (**C**) Median fluorescence intensity (MFI) of MeX04 in microglia. (**D**) MFI of CD45 on microglia. (**E**) % CD11c+ microglia of total microglia. (**F**) % MHC-II+ microglia of total microglia. (**G**) MFI of P2Y12 on microglia. Bars are plotted as mean + SEM. n=5–12. *p<0.05, **p<0.01, ***p<0.001 by two-way ANOVA with Sidak's multiple comparisons post hoc test.

The online version of this article includes the following source data and figure supplement(s) for figure 4:

**Source data 1.** Source data associated with *Figure 4*.

**Figure supplement 1.** Validating MeX04 to measure Aβ-containing microglia.

bioparticles, or myelin debris (*Figure 5—figure supplement 1A–D*); degradation of Aβ (*Figure 5—figure supplement 1E*); or lysosomal hydrolysis measured by the ability to cleave the quencher molecule from DQ-BSA (*Figure 5—figure supplement 1F*); transcriptional response to LPS, IL-4, or apoptotic neurons (*Figure 5—figure supplement 1G–I*); cytokine secretion in response to LPS or IL-4 (*Figure 5—figure supplement 2*) or NLRP3 inflammasome activation by silica particles (*Figure 5—figure supplement 1J and K*). We also addressed the possibility of compensation in adult knockout mice by using siRNA to knock down *Cst7* in the BV-2 microglia cell line and found that, despite 70% reduction in gene expression (*Figure 5—figure supplement 3A*), *Cst7* knockdown had no effect on IL-6 secretion (*Figure 5—figure supplement 3B*) or transcriptional changes response to LPS or IL-4 (*Figure 5—figure supplement 3C–E*).

Therefore, in order to investigate CF function *in vitro* in the context of the disease-associated marked induction of microglial *Cst7* expression (see above), we developed a model system using primary microglia isolated from 12-month *App*^*NL-G-F*^ mice and cultured them for 3 days *in vitro* (*Figure 5A*). We found that microglia isolated from *App*^*NL-G-F*^ mice then cultured for 3 days express

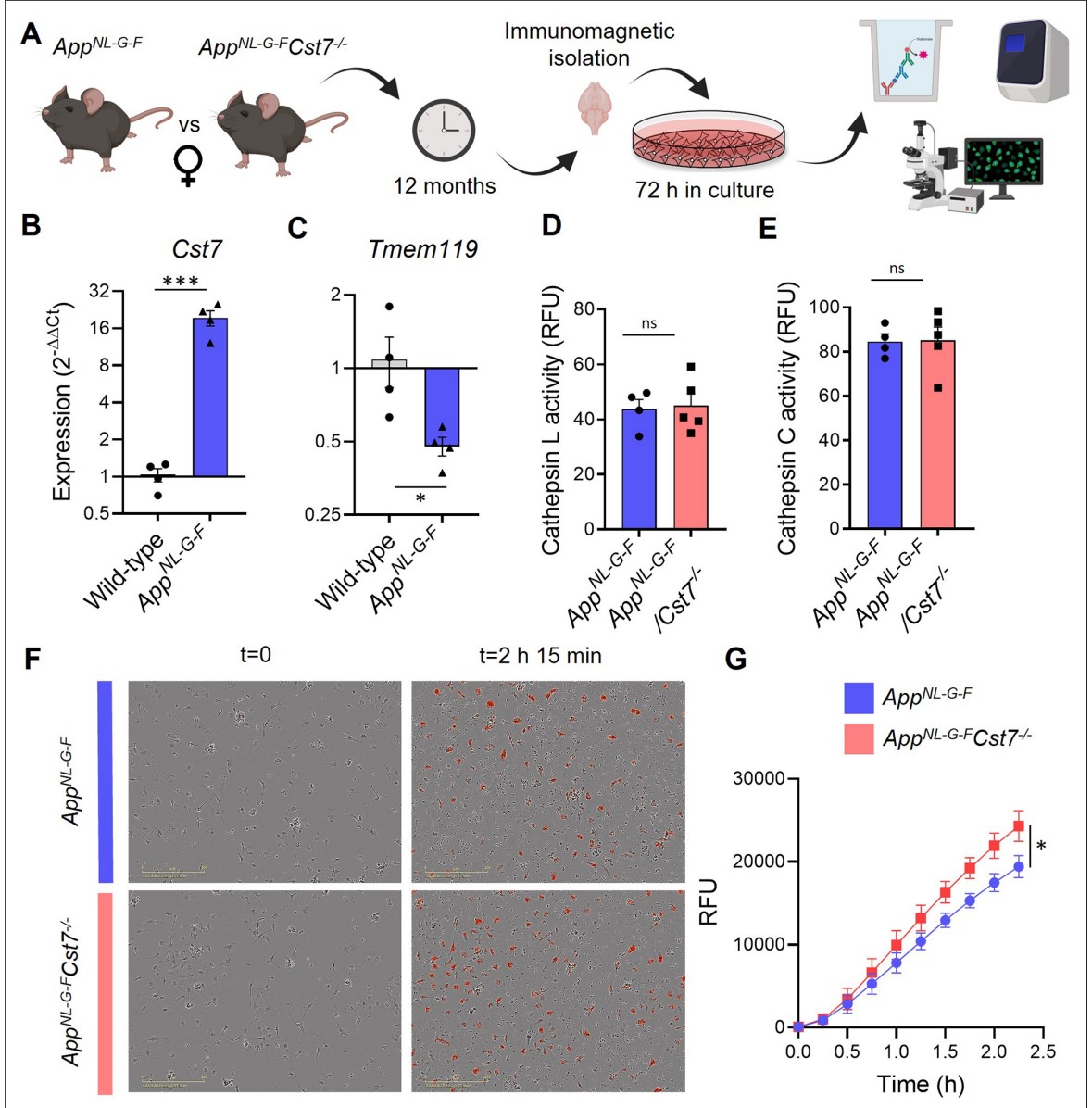

**Figure 5.** *Cst7* negatively regulates phagocytosis in microglia from female *App^NL-G-F* mice. (**A**) Study design schematic. (**B–C**) qPCR data for *Cst7* (**B**) and *Tmem119* (**C**) RNA expression from female wild-type and *App^NL-G-F* microglia isolated by CD11b beads and cultured for 3 days *in vitro*. Bars represent mean fold change calculated by ΔΔCt. n=4. *p<0.05, ***p<0.001 by two-way t-test. (**D–E**) Cathepsin L (**D**) and C (**E**) activity from female *App^NL-G-F*Cst7^+/+* (blue) and *App^NL-G-F*Cst7^-/-* (red) microglia measured by probe-based assay. Bars represent mean relative fluorescence units (RFU) + SEM. n=4–5. (**F**) Example images from myelin phagocytosis assays with female *App^NL-G-F*Cst7^+/+* (blue) and *App^NL-G-F*Cst7^-/-* (red) microglia taken immediately before (t=0) and 2 hr 45 min after (t=2 hr 45 min) addition of pHrodo-tagged myelin. (**G**) Quantification of (**F**). Data shown are mean RFU (calculated as total integrated intensity of pHrodo Red normalised to cell confluence) ± SEM. n=4. *p<0.05 calculated by unpaired t-test on area under curve (AUC) of data.

The online version of this article includes the following source data and figure supplement(s) for figure 5:

**Source data 1.** Source data associated with *Figure 5*.

**Figure supplement 1.** Investigating Cst7 knockout *in vitro*.

**Figure supplement 2.** Investigating Cst7 knockout on microglial cytokine secretion.

**Figure supplement 3.** Investigating Cst7 knockdown in BV-2 cells vitro.

**Figure supplement 4.** Investigating Cst7 knockout from App^NL-G-F* mice *in vitro*.

20-fold more *Cst7* than from age-matched wild-type counterparts (*Figure 5B*). We also found that these *App*^NL-G-F microglia express two-fold less *Tmem119*, an established microglial homeostatic gene, than wild-type cells (*Figure 5C*). These data suggest that, despite removal from a disease brain and subsequent culture *in vitro*, 12-month-old *App*^NL-G-F microglia retain key elements of their *in vivo* disease transcriptional identity.

We then used this validated system to compare functional properties of microglia isolated from female *App*^NL-G-F*Cst7*^-/- and *App*^NL-G-F*Cst7*^+/+ mice. First, we showed that *Cst7* knockout had no effect on isolation yield (*Figure 5—figure supplement 4A*) and observed no differences in inflammatory cytokine secretion in response to LPS or NLRP3 activation stimulus silica (*Figure 5—figure supplement 4B–D*), as predicted from our previous RNASeq analysis in female mice. Next, we tested whether *Cst7* plays a role in endolysosomal processes such as phagocytosis or lysosomal degradation. We used a pulse-chase live-imaging assay and found that *Cst7* deletion does not affect degradation of $A\beta_{1-42}$ (*Figure 5—figure supplement 4E*), a process dependent on cathepsins as it was inhibited by broad-spectrum inhibitors K777 and Ca074-Me (*Figure 5—figure supplement 4F*). We also showed that *Cst7* deletion does not affect intracellular activity of cathepsins L and C (*Figure 5D and E*), targets of cystatin-F (*Hamilton et al., 2008*), or microglial lysosomal hydrolysis (*Figure 5—figure supplement 4G*). These data would suggest that the increased microglial Aβ burden *in vivo* observed in *Figure 4A* is not due to decreased degradative capacity and may be due increased phagocytosis. To test this, we performed uptake/phagocytosis assays on microglia isolated from 12-month female *App*^NL-G-F*Cst7*^-/- vs. *App*^NL-G-F*Cst7*^+/+. To remove the confounding factor that female *App*^NL-G-F*Cst7*^-/- will already contain more Aβ (*Figure 4B and C*) and this may influence further Aβ uptake, we used myelin debris tagged with the pH-sensitive dye pHrodo Red. We found that *Cst7* deletion led to a significant increase in microglial phagocytosis of myelin debris (*Figure 5F and G*). We also tested uptake with fluorescently tagged $A\beta_{1-42}$ and found a trend towards increased uptake of $A\beta_{1-42}$ (*Figure 5—figure supplement 4H*).

Together, these data suggest that, in females, *Cst7*/CF negatively regulates phagocytosis but does not affect lysosomal proteolysis or PRR-driven inflammatory cytokine production in microglia from the *App*^NL-G-F model of amyloid-driven AD.

## Effect of *Cst7* deletion on disease pathology

Having discovered that *Cst7* plays sex-dependent regulatory roles on microglial phenotype and function in the *App*^NL-G-F model of amyloid-driven AD, we sought to determine if this is associated with effects on disease pathology. Therefore, we studied the Aβ plaques in 12-month male and female *App*^NL-G-F*Cst7*^+/+ and *App*^NL-G-F*Cst7*^-/- mice across brain regions. We observed the greatest plaque burden in the cortex and dorsal subiculum followed by the hippocampus (*Figure 6A*) and, as expected, the cerebellum was largely devoid of plaque pathology. Next, we compared plaque burden within each area between *App*^NL-G-F*Cst7*^+/+ and *App*^NL-G-F*Cst7*^-/- male and female mice. In males, there was no effect of *Cst7* genotype on plaque burden in any region (*Figure 6B–F*). Surprisingly, despite an increased lysosomal burden (*Figure 3F*) and intramicroglial Aβ load (*Figure 4B*) in female *App*^NL-G-F*Cst7*^-/- mice, there was no decrease in amyloid plaque burden in any region. In fact, the reverse pattern was evident in some areas, most notably in the subiculum (*Figure 6D*). In order to further probe the increase in amyloid pathology in the subiculum, we used MeX04 labelling of the dense cores of Aβ plaques and counted the number of plaques in this region. We observed a significant although modest increase in plaque number specifically in the females (*Figure 6G*, quantification in *Figure 6H*) without any difference in average plaque size detected (*Figure 6—figure supplement 1A*). Together, these data suggest that *Cst7*/CF plays a negative regulatory role on microglial endolysosomal function in female AD mice and show that removal of this block and resultant increase in phagocytosis in *App*^NL-G-F*Cst7*^-/- females is associated with increased plaque pathology. Finally, we assessed whether *Cst7* deletion had any effect on gross synapse coverage using synaptophysin (Sy38) immunostaining and thioflavin S (ThioS) to localise plaques. We observed loss of Sy38 coverage around plaques (*Figure 6I*) and a small but significant decrease in coverage between *App*^NL-G-F/*Cst7*^-/- vs. *App*^NL-G-F brains only in females (*Figure 6J*). This reflects the effect observed with plaque coverage above suggesting the increased plaque burden in *Cst7*^-/- female mice may lead to increased synapse loss.

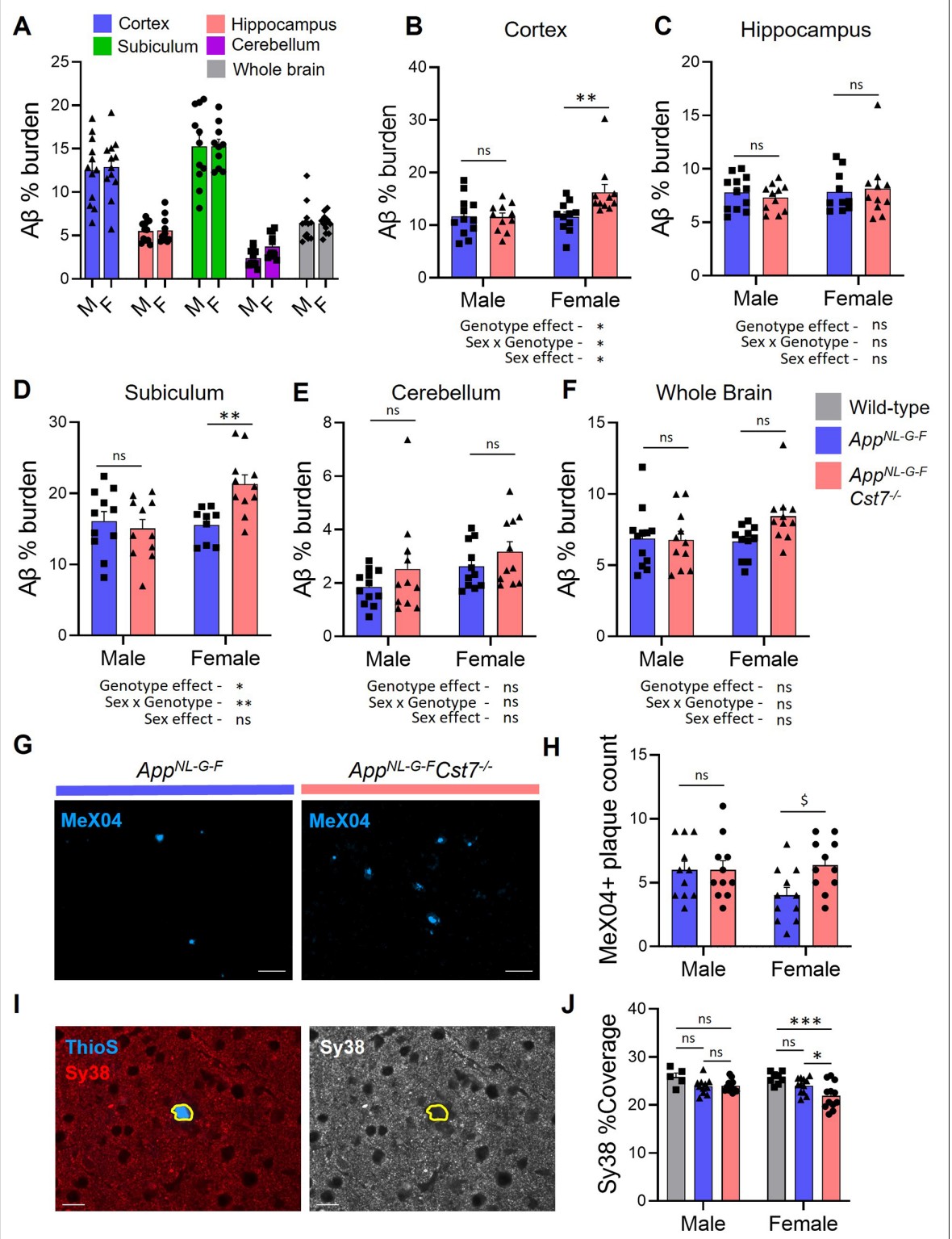

**Figure 6.** Investigating the effect of *Cst7* deletion on disease pathology. (**A**) % Aβ coverage measured by 6E10 3,3'-diaminobenzidine (DAB) staining in cortex (blue), hippocampus (red), subiculum (green), cerebellum (magenta), and whole brain (grey) of male and female *App*^NL-G-F^ brains. (**B–F**) % Aβ coverage in the cortex (**B**), hippocampus (**C**), subiculum (**D**), cerebellum (**E**), and whole brain (**F**) of male and female *App*^NL-G-F^*Cst7*^+/+^ (blue) and *App*^NL-G-F^*Cst7*^-/-^ (red) mice. Points are measured by thresholding of whole region area in sagittal section. Bars represent mean + SEM % coverage. n=10–12. *p<0.05, **p<0.01 calculated by two-way ANOVA with Sidak's multiple comparisons post hoc test. (**G**) Example images from subicula of female *App*^NL-G-F^*Cst7*^+/+^ (left) and *App*^NL-G-F^*Cst7*^-/-^ (right) mice stained with MeX04 (blue). Scale bars are 50 μm. (**H**) Quantification of (**G**). MeX04+ plaque count

*Figure 6 continued on next page*

*Figure 6 continued*

in the subicula of female *App^NL-G-F^Cst7^+/+^* (blue) and *App^NL-G-F^Cst7^-/-^* (red) mice. Points are number of plaques from a single field of view for each mouse. Bars are mean plaque count + SEM. n=10–12. $p<0.05 calculated by two-way Mann-Whitney test. (**I**) Example image of a plaque with Thio S (blue) and synapses stained with synaptophysin (red) in *App^NL-G-F^* brain. Plaque border is highlighted in yellow. (**J**) Quantification of (**I**) measuring coverage of synaptophysin. Bars represent mean + SEM % coverage. n=5–12. *p<0.05 calculated by two-way ANOVA with Tukey's multiple comparisons post hoc test.

The online version of this article includes the following source data and figure supplement(s) for figure 6:

**Source data 1.** Source data associated with *Figure 6*.

**Figure supplement 1.** Investigating the role of Cst7 in plaque burden.

## Discussion

Changes in microglial cells have been heavily implicated in the pathogenesis of AD, particularly supported by genetics studies, where expression of over 50% of risk genes is enriched in microglia (*Hansen et al., 2018*). Notably, a transcriptomic profile has emerged involving *Trem2*-dependent downregulation of homeostatic genes and upregulation of a signature characterised by genes involved in phagocytosis, lipid handling, and endolysosomal transport (*Deczkowska et al., 2018*). One of the most robustly upregulated of these genes, *Cst7* (CF), is believed to be involved in protease inhibition. However, the cellular functions controlled by *Cst7*/CF in microglia and whether Cst7/CF has a disease-modifying role in AD-related CNS proteinopathy has not been explored previously. Here, we reveal that *Cst7*/CF plays a sexually dimorphic role in microglia in an amyloid-driven AD model, acting as a restraint on microglial endolysosomal activity and phagocytosis specifically in female mice, that when absent results in subtle aggravation of Aβ pathology and synapse loss.

Since their initial description in 2017, the DAM/MGnD/ARM gene signature has been extensively investigated with over 4000 citations between the three papers describing them in 5 years. However, beyond GO analysis, there is surprisingly little understanding of what many of the hallmark genes comprising this signature functionally do in the context of neurodegenerative disease. This is the first study to our knowledge to test the functional role of *Cst7*, one of the most consistently replicated markers of this cell state transcriptomic signature, in β-amyloid-driven pathology. Perhaps counterintuitively, *Cst7* (a reported cysteine protease inhibitor) is robustly upregulated in microglia alongside concomitant upregulation of cysteine proteases such as cathepsins D, B, L, and Z. In fact, such internal regulation systems are not uncommon in inflammation biology, where proinflammatory mediators such as IL-1β and IL-18 are negatively regulated by co-expressed IL-1 receptor antagonist and IL-18 binding protein, respectively (*Hurme and Santtila, 1998*; *Dinarello et al., 2013*). We therefore hypothesise that *Cst7*/CF is upregulated along with numerous cathepsin genes in order to prevent potentially harmful protease activity (*Turk et al., 2012*). In this study, we show that *Cst7*/CF knockout triggers an increase in microglial lysosomal activity and amyloid uptake. Intriguingly, this did not seem to depend on overactive intracellular cathepsin activity, which intuitively would have been expected to increase with the lack of negative regulator. It is possible that CF deficiency does in fact lead to an increase in cathepsin activity, but CF itself is secreted and only impairs secreted cathepsins which are responsible for promoting phagocytosis. Indeed, studies have suggested that CF can be secreted and that there is a role for secreted cathepsins in phagocytosis (*Liuzzo et al., 1999*; *Hamilton et al., 2008*). This is also consistent with the observation that exogenously applied cystatin C can block phagocytosis in human polymorphonuclear neutrophils (*Leung-Tack et al., 1990*).

Another key implication from this study is that of sexual dimorphism in disease. Here, we show stark differences in the effect of *Cst7* deletion between males and females at both the gene and protein level. Although there is a relative paucity of studies investigating male *vs.* female microglia in homeostasis and disease, numerous differences have been identified (*Lynch, 2022*). In general, male microglia are believed to have higher baseline activity in processes such as inflammation, antigen presentation, and phagocytosis; whereas female microglia are more associated with neuroprotection and the DAM/ARM/MGnD signature in ageing and disease (*Guneykaya et al., 2018*; *Kodama and Gan, 2019*; *Sala Frigerio et al., 2019*; *Villa et al., 2019*; *Guillot-Sestier et al., 2021*). These differences in baseline properties are only partially dependent on hormonal signalling as masculinisation of females by ovariectomy and hormone replacement only partially recapitulated male microglia (*Villa et al., 2018*). Indeed, sexual dimorphism of microglia in adults appears a cell-intrinsic

phenomenon as female microglia transplanted in male brains retain their transcriptional profile and reduce infarct size in males after ischemic stroke (*Villa et al., 2018*). Interestingly, we generally do not observe marked sexual dimorphism in microglia in wild-type or $App^{NL-G-F}$ mice. However, DAM/ARM/MGnD gene expression was higher in female *vs.* male microglia and lysosomal activity was markedly higher in male *vs.* female brains measured by LAMP2 staining, which is consistent with the literature (*Guneykaya et al., 2018*; *Sala Frigerio et al., 2019*). Importantly, we observed a substantial interaction effect of *Cst7* deletion and sex, whereby removal of the *Cst7* gene led to an 'unlocking' of sexual dimorphism in our cohort. This is most clearly demonstrated by DEGs between male *vs.* female microglia rising from 33 in $App^{NL-G-F}Cst7^{+/+}$ mice to 240 in $App^{NL-G-F}Cst7^{-/-}$ mice. This sex-dependent interaction effect has previously been observed in studies investigating the cystatin/cathepsin system, where deletion of *Cst3* (cystatin C) led to protection in the EAE model of multiple sclerosis in females but not males (*Hoghooghi et al., 2020*). This effect appeared to be sensitive to hormones as ovariectomy or castration followed by testosterone or estrogen/progesterone administration effectively reversed the effect of *Cst3* knockout. The precise molecular mechanism underpinning this was not explored but could include differential post-translational modification. The inhibitor effect of cystatin C on phagocytosis is dependent on N-terminal truncation (*Leung-Tack et al., 1990*) whilst more recently an N-terminal processing event has been identified for CF (*Hamilton et al., 2008*). Understanding whether N-terminal processing events are sex-specific would be a useful step in determining how genetic deletion can have differential effects dependent on sex. Indeed, sex:-genetics interactions are not uncommon in neurodegenerative disease, with various risk variants posing greater or lesser effect on risk in men or women (*Gamache et al., 2020*). One example of this interaction is *APOE*, in which the presence of the E4 allele confers a greater AD risk in women than in men (*Altmann et al., 2014*).

## Ideas and speculation

Here, we observed that *Cst7*/CF played a role in microglial amyloid uptake and endolysosomal gene expression only in females. Surprisingly, this did not lead to a decrease in Aβ plaque pathology as might be expected but an increase, although of modest degree, in specific brain areas. This discovery raises the important question of the relationship between microglial phagocytosis and plaques. It is generally considered that an increase in microglial phagocytosis should lead to reduction in plaque burden and benefit in disease (*Heneka et al., 2013*; *Tejera et al., 2019*). However, recent studies have emerged to suggest that phagocytosing microglia may in fact act to 'build' plaques rather than dismantle them. For example, microglia lacking TAM receptors Axl and Mer fail to take up Aβ which leads to an overall decrease in dense-core plaque formation (*Huang et al., 2021*), microglia can 'seed' plaques from disease tissue into engrafted, non-affected regions (*d'Errico et al., 2022*), and either genetic or pharmacological depletion of microglia leads to a reduction in plaque burden/intensity accompanied by increase in cerebral amyloid angiopathy (*Spangenberg et al., 2019*; *Kiani Shabestari et al., 2022*) perhaps through redistributing Aβ. Interestingly, only female mice were used in two of these studies (*Huang et al., 2021*; *d'Errico et al., 2022*), which parallels our sex-specific findings with *Cst7*. In the above studies, microglial depletion/inactivation leading to reduced parenchymal plaque burden is generally detrimental in disease. In contrast, we find indications in our study that the increased microglial phagocytosis and concomitant increase in plaque burden in $Cst7^{-/-}$ females may be detrimental. This is most likely due to direct effect of increased plaque burden leading to increased synapse loss. However, we cannot confirm whether this apparent synapse loss is a result of increased microglial uptake (*Hong et al., 2016*) or direct synaptotoxicity (*Tzioras et al., 2023b*). We speculate that *Cst7*/CF may contribute to restraining plaque formation and protecting against synapse loss by influencing microglial endolysosomal function in a dose-dependent manner.

We were also surprised that there was no change in plaque burden in males, where microglia had a reduction in inflammatory profile. Studies have suggested that reducing inflammatory cytokine secretion may reduce plaque formation and increase Aβ uptake (*Heneka et al., 2013*; *Tejera et al., 2019*). These effects may be linked specifically to the NLRP3-caspase-1-ASC-IL-1β axis and that a more general reduction in inflammatory genes is not capable of phenocopying $Nlrp3^{-/-}$ mice. Additionally, we observe a relatively mild reduction in inflammatory genes in male $App^{NL-G-F}Cst7^{-/-}$ mice with fold changes mostly below 2 *vs.* $App^{NL-G-F}Cst7^{+/+}$ controls. We speculate that either complete genetic deletion or strong inhibition with chemical compounds is required to yield pro-phagocytic effects.

We also note in our study that male *Cst7⁻/⁻* microglia had reduced expression of lysosomal proteins, possibly counteracting the reduction in inflammatory gene expression.

## Limitations

Although these data provide novel insight into the sex-dependent role of microglial gene *Cst7* in AD, it is important to acknowledge some caveats of the study that point to potential future investigations into this complex biological phenomenon. Here, we investigated CF function solely in mouse. While human data on microglial CF is limited, studies have shown that (where detected) CF is expressed predominantly in myeloid cells (*Zhang et al., 2016*; *Mathys et al., 2019*; *Schirmer et al., 2019*; *Srinivasan et al., 2020*) and identified CF expression in microglia around plaques in AD (*Ofengeim et al., 2017*) and *CST7* enrichment in AD-associated microglia clusters by single nucleus RNA sequencing (snRNASeq) (*Gerrits et al., 2021*). CST7 protein has also been detected as higher levels predicted slower tau accumulation and cognitive decline with a significant sex interaction effect (*Pereira et al., 2022*). However, CF/*CST7* has also remained undetected in similar immunostaining and snRNASeq studies (*Nuvolone et al., 2017*; *Zhou et al., 2020*). While the reason for this discrepancy is unclear, it may be partially due to sex. Indeed, there was a greater ratio samples from female brains in *Gerrits et al., 2021* than in *Zhou et al., 2020*, as would be expected if *CST7* plays a more important role in female microglia. These data further demonstrate the importance of stratifying studies by sex. Additionally, while mechanistic studies investigating CF/*CST7* in human microglia differentiated from inducible pluripotent stem cells would be valuable, our discovery that CF function is revealed in mouse microglia only when cells are taken from disease context would suggest more complex models are required. Indeed, a recent study utilised overexpression of TREM2 in human microglial cells to show that driving *CST7* expression led to inhibition of phagocytosis in a *CST7*-dependent manner (*Popescu et al., 2023*). These data suggest the functional role of CF is not species-specific and provide important translational evidence. Finally, while this study details the discovery of the mechanistic role of CF/*Cst7* in an amyloid-driven AD model, the precise mechanism(s) by which *Cst7* deletion affects microglia only in females and the biological reason for the accelerated DAM/MGnD/ARM programme in females now demonstrated in multiple studies remains unknown. This phenomenon may go some way to explaining the sexual dimorphism we observe in this study, that is, the effect of *Cst7* deletion at 12 months in females may be mirrored in males at later disease stages. However, although we do observe increased expression of sex-regulated DAM genes such as *Gpnmb* and *Spp1*, we do not find differential expression of *Cst7* itself in females *vs.* males or most other genes in *App^NL-G-F^*. Additionally, we find a qualitatively different pattern of altered gene expression from *Cst7* deletion in female *vs.* male *App^NL-G-F^* mice rather than similar effects of differing magnitude. This suggests that that a sex-intrinsic mechanism rather than a simple 'lagging' in males is a more likely explanation underpinning *Cst7* deletion sex-dependent effects. Future work should investigate this further with ovariectomy/hormone replacement and transplantation studies, such as those performed by *Villa et al., 2018*.

In summary, our data provide key mechanistic insight into the role of one of the most robustly upregulated genes in disease-reactive microglia. These data suggest that *Cst7*/CF regulates some key aspects of microglial function, in a sex-dependent manner, and that these are associated with pathology-influencing effects in an amyloid-driven AD model that also manifest differently in males and females. We hypothesise that *Cst7*/CF plays a part in an internal regulatory system that balances the two crucial processes of phagocytosis and inflammatory signalling, which are sexually dimorphic processes. More broadly, and in view of the poorly understood functions of many other recently described hallmark microglial disease-related state mediators, it is imperative to consider interactions with sex when investigating their cellular and disease roles.

# Materials and methods

## Key resources table

| Reagent type (species) or resource | Designation | Source or reference | Identifiers | Additional information |
|---|---|---|---|---|
| Genetic reagent (*Mus musculus*) | *App^NL-G-F^* | DOI:10.1038/nn.3697 | | |

*Continued on next page*

*Continued*

| Reagent type (species) or resource | Designation | Source or reference | Identifiers | Additional information |
|---|---|---|---|---|
| Genetic reagent (*Mus musculus*) | *Cst7$^{-/-}$* | DOI:10.1016/j.immuni.2016.03.003 | | |
| Cell line (*Mus musculus*) | BV2 | Collaborator | RRID: CVCL_0182 | Authenticated (STR profiling) |
| Cell line (*Homo sapiens*) | SH-SY5Y | Collaborator | RRID: CVCL_0019 | Authenticated (STR profiling) |
| Biological sample (*Homo sapiens*) | Synaptoneurosome from AD brain | DOI:10.1016/j.xcrm.2023.101175 | | AMREC (approval number 15-HV-016) |
| Biological sample (*Mus musculus*) | Purified myelin | This paper | | See Materials and methods 'Myelin purification and pHrodo tagging' |
| Antibody | Anti-Ly6C- Alexa Fluor 488 (Rat monoclonal) | BioLegend | Cat# 128021 | FACS 1:500 |
| Antibody | Anti-P2Y12 – PE (Rat monoclonal) | BioLegend | Cat# 848003 | FACS 1:200 |
| Antibody | Anti-MHCII – PEDazzle594 (Rat monoclonal) | BioLegend | Cat# 107647 | FACS 1:200 |
| Antibody | Anti-CD45 – PE-Cy7 (Rat monoclonal) | BioLegend | Cat# 103113 | FACS 1:200 |
| Antibody | Anti- CD11c – APC (Armenian Hamster monoclonal) | BioLegend | Cat# 117309 | FACS 1:200 |
| Antibody | Anti- CD11b – BV711 (Rat monoclonal) | BioLegend | Cat# 848003 | FACS 1:50 |
| Antibody | Anti-IBA1 (Rabbit polyclonal) | Wako | Cat# 019-19741 | IF 1:2000 |
| Antibody | Anti- Aβ (6E10) (Mouse monoclonal) | BioLegend | Cat# 803001 | IF/DAB 1:1000 |
| Antibody | Anti-LAMP2 (Rat monoclonal) | BioLegend | Cat# 108501 | IF 1:200 |
| Antibody | Anti-Synaptophysin (Sy38) (Mouse monoclonal) | Abcam | ab8049 | IF 1:200 |
| Antibody | Anti-rabbit IgG Alexa Fluor 488 (Goat polyclonal) | ThermoFisher | Cat# A-11008 | IF 1:500 |
| Antibody | Anti-mouse IgG Alexa Fluor 555 (Donkey polyclonal) | ThermoFisher | Cat# A-31570 | IF 1:500 |
| Antibody | Anti-rat IgG Alexa Fluor 647 (Goat polyclonal) | ThermoFisher | Cat# A-21247 | IF 1:500 |
| Antibody | Anti-mouse Biotinylated | Vector Labs | Cat# ba-9200 | DAB 1:100 |
| Chemical compound, drug | Methoxy X04 | Tocris | Cat# 4920 | |
| Chemical compound, drug | Cathepsin L probe | Bachem | Cat# 4003379 | Z-Phe-Arg-AMC |
| Chemical compound, drug | Cathepsin C probe | Bachem | Cat# 4003759 | H-Gly-Phe-AMC |
| Sequence-based reagent | Cst7_F | This paper | PCR primers | ACCAATAACCCAGGAGTGCTTA |
| Sequence-based reagent | Cst7_R | This paper | PCR primers | TGACCCAGACTTCAGAGTAGCA |

*Continued on next page*

*Continued*

| Reagent type (species) or resource | Designation | Source or reference | Identifiers | Additional information |
|---|---|---|---|---|
| Sequence-based reagent | Arg1_F | This paper | PCR primers | GGAGACCACAGTCTGGCAGTTGGA |
| Sequence-based reagent | Arg1_R | This paper | PCR primers | GGACACAGGTTGCCCATGCAGA |
| Sequence-based reagent | Il1b_F | This paper | PCR primers | CGACAAAATACCTGTGGCCTTGGGC |
| Sequence-based reagent | Il1b_R | This paper | PCR primers | TGCTTGGGATCCACACTCTCCAGC |
| Sequence-based reagent | Trem2_F | This paper | PCR primers | CTGCTGATCACAGCCCTGTCCCAA |
| Sequence-based reagent | Trem2_R | This paper | PCR primers | CCCCCAGTGCTTCAAGGCGTCATA |
| Sequence-based reagent | Gapdh_F | This paper | PCR primers | TGCATCCACTGGTGCTGCCAA |
| Sequence-based reagent | Gapdh_R | This paper | PCR primers | ACTTGGCAGGTTTCTCCAGGCG |
| Sequence-based reagent | Lilrb4a_F | This paper | PCR primers | ATGGGCACAAAAAGAAGGCTAA |
| Sequence-based reagent | Lilrb4a_R | This paper | PCR primers | GGCATAGGTTACATCCTGGGTC |
| Sequence-based reagent | Ndufv1_F | This paper | PCR primers | ATTTTCTCGGCGGGTTGGTT |
| Sequence-based reagent | Ndufv1_R | This paper | PCR primers | CACCTTTCAGCCTCCAGTCA |
| Commercial assay or kit | DuoSet ELISA – IL-1β | R&D Systems | Cat# DY401 | |
| Commercial assay or kit | DuoSet ELISA – IL-6 | R&D Systems | Cat# DY406 | |
| Commercial assay or kit | DuoSet ELISA – TNFα | R&D Systems | Cat# DY411 | |
| Commercial assay or kit | RNAscope 2.5 HD Duplex Assay | Biotechne | Cat# 322436 | |
| Commercial assay or kit | RNAscope probes – Cst7 | Biotechne | Cat# 498711-C2 | |
| Commercial assay or kit | RNAscope probes – Lilrb4a | Biotechne | Cat# 1260291-C2 | |

## Animals

*App^NL-G-F* mice (***Saito et al., 2014***) were provided by the RIKEN BRC through the National BioResource Project of the MEXT/AMED, Japan. *Cst7^-/-* mice (***Matthews et al., 2016***) were obtained from Colin Watts, University of Dundee, UK. C57Bl/6J mice used for myelin preparations were purchased from Charles River, UK. Animals were maintained under standard laboratory conditions: ambient temperatures of 21°C (±2°C), humidity of 40–50%, 12 hr light/dark cycle, *ad libitum* access to water, and standard rodent chow. Genotype groups were randomised during the study and experimenters were blinded to genotype during all experiments. All animal experiments were carried out in accordance with the United Kingdom Animals (Scientific Procedures) Act 1986 and approved by the Home Office and the local Animal Ethical Review Group, University of Edinburgh. Experimental design, analysis, and reporting followed the ARRIVE 2.0 guidelines (***Percie du Sert et al., 2020***). Genotyping was performed using optimised assays from Transnetyx. Methoxy X-04 (MeX04, BioTechne) was reconstituted in DMSO at 10 mg/mL before diluting to 0.33 mg/mL in 6.67% Cremaphor EL (Fluka), 90% PBS

**Table 1.** Antibodies used for flow cytometry.

| Fluorochrome | Antigen | Clone | Lot | Cat number | Stock concentration (mg/mL) | Dilution |
|---|---|---|---|---|---|---|
| Alexa Fluor 488 | Ly6C | HK1.4 | B248739 | 128021 | 0.5 | 500 |
| PE | P2Y12 | S16007D | B298459 | 848003 | 0.2 | 200 |
| PEDazzle594 | MHCII | M5/114.15.2 | B216062 | 107647 | 0.2 | 200 |
| PE-Cy7 | CD45 | 30-F11 | B271123 | 103113 | 0.2 | 200 |
| APC | CD11c | N418 | B280313 | 117309 | 0.2 | 200 |
| BV711 | CD11b | M1/70 | B305911 | 101241 | 0.005 | 50 |

(Merck). MeX04 was administered by intraperitoneal injection at 10 mg/kg 2.5 hr before animals were terminated.

## Microglial FACS isolation

Microglia were isolated by enzymatic digestion and FACS as follows. Brains from 12-month-old male and female $App^{Wt/Wt}Cst7^{+/+}$, $App^{NL-G-F}Cst7^{+/+}$, and $App^{NL-G-F}Cst7^{-/-}$ mice were isolated by terminally anaesthetising with 3% isoflurane (33.3% $O_2$ and 66.6% $N_2O$) and transcardial perfusion with ice-cold DEPC-treated 0.9% NaCl, 0.4% trisodium citrate. Brains were immediately separated down the midline then the right hemisphere placed into ice-cold HBSS (ThermoFisher) and minced using a 22A scalpel. Minced hemi-brains were then centrifuged (300 × $g$, 2 min) and digested using the MACS Neural Dissociation Kit (Miltenyi) according to the manufacturer's instructions. Briefly, brain tissue was incubated in enzyme P (50 µL/hemi-brain) diluted in buffer X (1900 µL/hemi-brain) for 15 min at 37°C under gentle rotation before addition of enzyme A (10 µL/hemi-brain) in buffer Y (20 µL/hemi-brain) and further incubation for 20 min at 37°C under gentle rotation. Following digestion, tissue was dissociated mechanically using a Dounce homogeniser (loose pestle, 20 passes) on ice and centrifuged (400 × $g$, 5 min at 4°C). To remove myelin, tissue was resuspended in 35% isotonic Percoll (GE Healthcare) in HBSS overlaid with 1× HBSS and centrifuged (800 × $g$, 30 min, 4°C, no brake). Following centrifugation, the supernatant and myelin layers were discarded and the pellet resuspended in 100 µL FACS buffer (PBS, 0.1% low endotoxin bovine serum albumin [BSA, Merck], 25 mM HEPES) and transferred to V-bottomed 96-well plates (ThermoFisher). Cell suspensions were treated with anti-mouse-CD16/32 to block Fc receptors (BioLegend, 5 µg/mL, 20 min at 4°C) before washing and transferring to PBS and treating with Zombie NIR (BioLegend, 1:100, 15 min at room temperature [RT]) to label dead cells. Next, cells were stained with antibodies (*Table 1*) to identify microglia and incubated for 20 min at 4°C. Antibodies (BioLegend) used were as follows.

Following incubation, cells were washed with FACS buffer, resuspended in 500 µL FACS buffer and transferred to 5 mL round-bottom FACS tubes through cell strainer caps (BD). Cells were sorted through a 100 µm nozzle on a FACS Aria II (BD) at the QMRI Flow Cytometry Cell Sorting Facility, University of Edinburgh with the following strategy (*Figure 2—figure supplement 1*). Debris were eliminated using forward scatter (area) vs. side scatter (area), singlets were isolated using forward scatter (area) vs. forward scatter (height), dead cells were eliminated with Zombie NIR, monocytes were eliminated by gating out Ly6C+ cells, and microglia were defined as CD11b+/CD45+. All antibodies were validated using appropriate fluorescence minus one controls. Microglia were sorted directly into 500 µL RLT buffer (QIAGEN) and 20,000 CD11b+/CD45+ events were collected for downstream analysis using FlowJo software.

## RNA purification, QC, and sequencing

RNA was purified using RNEasy Plus Micro kits (QIAGEN) according to the manufacturer's instructions. RNA was quantified and QC'd using a 4200 TapeStation System (Agilent) with a High Sensitivity RNA ScreenTape Assay according to the manufacturer's instructions. For *in vivo* experiments, RNA was pooled from three to four animals within the same experimental group. RNA within pools was equally distributed in amount between the animals. One ng of total RNA was taken forward for low-input RNA sequencing (Cambridge Genomics). Briefly, cDNA was generated using TakaraBio SMART-Seq v4

Ultra Low Input RNA kit before input into the Illumina Nextera XT library prep. Pooled libraries were sequenced on the NextSeq 500 (Illumina) using the 75 cycle High Output sequencing run kit, spiked with 5% PhiX at a depth of 30 million reads per sample.

## RNASeq analysis

Reads were mapped to the mouse primary genome assembly GRCm39, Ensembl release 104 (*Cunningham et al., 2022*) using STAR version 2.7.9a (*Dobin et al., 2013*), and tables of per-gene read counts were summarised using featureCounts version 2.0.2 (*Liao et al., 2014*). Differential expression analysis was performed using DESeq2 (R package version 1.30.1) (*Love et al., 2014*), using an adjusted p-value cut-off of 0.05 to identify genes differentially expressed between conditions. Comparisons carried out were male (M) $App^{Wt/Wt}Cst7^{+/+}$ vs. M $App^{NL-G-F}Cst7^{+/+}$, female (F) $App^{Wt/Wt}Cst7^{+/+}$ vs. F $App^{NL-G-F}Cst7^{+/+}$, M $App^{NL-G-F}Cst7^{+/+}$ vs. M $App^{NL-G-F}Cst7^{-/-}$, F $App^{NL-G-F}Cst7^{+/+}$ vs. F $App^{NL-G-F}Cst7^{-/-}$, M $App^{NL-G-F}Cst7^{+/+}$ vs. F $App^{NL-G-F}Cst7^{+/+}$, M $App^{NL-G-F}Cst7^{-/-}$ vs. F $App^{NL-G-F}Cst7^{-/-}$. For each comparison GO enrichment analysis was performed to provide insight into the biological pathways and processes affected; GO enrichment analysis was performed using topGO (*Alexa et al., 2006*) (R package version 2.42.0). Significantly upregulated and downregulated gene sets were compared against annotated GO terms from the categories of 'biological process'. Selected statistically significant GO:BP terms were then visualised using GraphPad prism v9.

## qPCR

Cells were seeded at 75,000 cells/well in a 24-well plate 7 days before stimulation (primary microglia) or 250,000 cells/well in a 24-well plate 18 hr before stimulation (BV-2 cells). On the day of the assay, microglia were treated with vehicle (PBS, 24 hr, Merck), LPS (100 ng/mL, 24 hr, Merck), IL-4 (20 ng/mL, 24 hr, R&D Systems) or apoptotic SH-SY5Y neuroblastoma cells (2:1 ratio neurons:microglia, 24 hr) then lysed in RLT buffer and RNA isolated according to the manufacturer's instructions. RNA (100–500 ng) was converted to cDNA using SuperScript IV Reverse Transcriptase (ThermoFisher) according to the manufacturer's instructions. qPCR was performed using PowerUp SYBR Green PCR Master Mix (ThermoFisher) in 384-well format using an qTOWER³84 Real-time PCR machine (Analytik Jena). For BV-2 cells, 10 ng cDNA (assuming 100% RNA to cDNA conversion) was loaded per well with 5 pmol primer/well in triplicate in a total volume of 10 µL. For primary microglia experiments, 0.5 ng cDNA was loaded per well. Data were normalised to the expression of the housekeeping gene *Gapdh* or mean *Gapdh* and *Ndufv1* and were analysed using the ΔΔCt method. Primers used are detailed in *Table 2*.

## Microglial bead-based isolation for *in vitro* studies

Primary adult mouse microglia were isolated and cultured as described previously (*Grabert and McColl, 2018*). Brains were isolated by terminally anaesthetising with 3% isoflurane (33.3% $O_2$ and 66.6% $N_2O$) and transcardial perfusion with ice-cold 0.9% NaCl. Brains were immediately placed into ice-cold HBSS (ThermoFisher) and minced using a 22A scalpel before centrifugation (300 × *g*, 2 min) and digestion using the MACS Neural Dissociation Kit (Miltenyi) according to the manufacturer's instructions. Briefly, brain tissue was incubated in enzyme P (50 µL/brain) diluted in buffer X (1900 µL/brain) for 15 min at 37°C under gentle rotation before addition of enzyme A (10 µL/brain) in buffer

**Table 2.** Primers used for qPCR.

| Gene | Forward | Reverse |
|---|---|---|
| *Cst7* | ACCAATAACCCAGGAGTGCTTA | TGACCCAGACTTCAGAGTAGCA |
| *Arg1* | GGAGACCACAGTCTGGCAGTTGGA | GGACACAGGTTGCCCATGCAGA |
| *Il1b* | CGACAAAATACCTGTGGCCTTGGGC | TGCTTGGGATCCACACTCTCCAGC |
| *Trem2* | CTGCTGATCACAGCCCTGTCCCAA | CCCCCAGTGCTTCAAGGCGTCATA |
| *Gapdh* | TGCATCCACTGGTGCTGCCAA | ACTTGGCAGGTTTCTCCAGGCG |
| *Lilrb4a* | ATGGGCACAAAAAGAAGGCTAA | GGCATAGGTTACATCCTGGGTC |
| *Ndufv1* | ATTTTCTCGGCGGGTTGGTT | CACCTTTCAGCCTCCAGTCA |

Y (20 µL/brain) and further incubation for 20 min at 37°C under gentle rotation. Following digestion, tissue was dissociated mechanically using a Dounce homogeniser (loose pestle, 20 passes) on ice and centrifuged (400 × $g$, 5 min at 4°C). To remove myelin, tissue was resuspended in 35% isotonic Percoll (GE Healthcare) overlaid with HBSS and centrifuged (800 × $g$, 40 min, 4°C). Following centrifugation, the supernatant and myelin layers were discarded and the pellet resuspended in MACS buffer (PBS, 0.5% low endotoxin BSA [Merck], 2 mM EDTA, 90 µL/brain). Anti-CD11b microbeads (Miltneyi) were added (10 µL/brain) and the suspension incubated for 15 min at 4°C before running through pre-rinsed (MACS buffer) LS columns attached to a magnet (Miltenyi). After washing with 12 mL MACS buffer, columns were removed from the magnet and cells retained (microglia) were flushed in 5 mL MACS buffer. For experiments investigating expression of RNASeq hit genes in non-diseased microglia, cells were immediately lysed in RLT buffer and RNA purified as above. For culture experiments, microglia were resuspended in Dulbecco's Modified Eagle's Medium/Nutrient Mixture F-12 (DMEM/F-12, ThermoFisher) supplemented with 100 U/mL penicillin and 100 µg/mL streptomycin (PenStrep, Merck), 10% heat-inactivated fetal bovine serum (FBS, ThermoFisher), 50 ng/mL rhTGFβ-1 (Miltenyi), 10 ng/µL mCSF1 (R&D Systems). Microglia were counted using a haemocytometer and plated out onto 24- or 96-well plates (Corning) coated with poly-L-lysine (Merck). Cells were cultured for 7 days with a half media change on day 3. For stimulation experiments, cells were stimulated with LPS, IL-4, or silica particles (US Silica) at concentrations and timepoints as described in each experiment in detail before supernatant was taken and assessed for IL-6, IL-1β, or TNF-α by DuoSet ELISA (R&D Systems) as per the manufacturer's instructions.

## Immortalised cell culture

Murine BV-2 microglial cells (**Blasi et al., 1990**) were cultured in DMEM (ThermoFisher), 10% FBS (ThermoFisher), and 1% PenStrep (Merck). Human SH-SY5Y neuroblastoma cells (ECACC 94030304) were cultured in DMEM/F-12 (ThermoFisher), 10% FBS (ThermoFisher), and 1% PenStrep (Merck). To induce apoptosis, SH-SY5Y cells were plated in a 10 cm dish at $4×10^6$ cells/dish and stimulated with UV light (4×15 W UV bulb, 30 hr) before incubation at 37°C for 18 hr. After incubation, cells were removed from dishes using Accutase (Merck) and adjusted to $4×10^6$ cells/mL for stimulation onto microglia. All cells were cultured at 37°C, 5% $CO_2$ in a tissue culture incubator (Panasonic) and handled in a Safe 2020 Class II tissue culture cabinet (ThermoFisher). Cell lines (BV-2 immortalized mouse microglia and SH-SY5Y human neuroblastoma) were obtained from collaborators. Identities of the cell lines were authenticated by STR profiling (American Type Culture Collection) and tested negative for mycoplasma (Mycoplasma Detection Kit, Lonza).

## BV-2 siRNA knockdown

Small interfering RNAs (siRNAs) (Silencer-Select, Ambion) targeting *Cst7* were used to induce gene knockdown in BV-2 cells. Cells were plated out at 250,000 cells/well in a 24-well plate 18 hr before knockdown. For transfection of each siRNA (*Cst7* and non-targeting control), a 1:1 ratio of Lipofectamine RNAiMAX (ThermoFisher) and the siRNA, both diluted in Opti-MEM (ThermoFisher), were added to desired cell supernatants, with 10 pmol siRNA and 1.5 µL Lipofectamine RNAiMAX used per well. After 24 hr, cells were stimulated with vehicle (PBS), LPS (100 ng/mL, 24 hr, Merck), or IL-4 (20 ng/mL, 24 hr, R&D Systems). Following stimulation, cells were lysed for RNA analysis and supernatants were analysed for IL-6 content by ELISA (DuoSet, R&D Systems) according to the manufacturer's instructions.

## Myelin purification and pHrodo tagging

Purified myelin was prepared from C57Bl/6J mouse brains. Brains were isolated, digested, and myelin layer fractionated using 35% percoll gradient as described above. Myelin was washed by dilution in HBSS and centrifuged at 400 × $g$ for 5 min at 4°C before suspending in 3 mL ice-cold 0.32 M sucrose solution containing protease inhibitor cocktail (Roche). Next, myelin was layered onto 3 mL ice-cold 0.85 M sucrose solution in a 10 mL ultracentrifuge tube (Beckman) before centrifugation at 75,000 × $g$ for 30 min at 4°C on a Beckman Ultracentrifuge with MLA-55 rotor (acceleration 7, deceleration 7). Myelin was collected from the 0.32 M:0.85 M interface and placed in 5 mL ice-cold ddH$_2$O in a new Beckman ultracentrifuge tube and vortexed before further centrifugation at 75,000 × $g$ for 15 min at 4°C (acceleration 9, deceleration 9). Myelin was then reconstituted in 5 mL ddH$_2$O and incubated on

ice for 10 min before further centrifugation at 12,000 × $g$ for 15 min at 4°C (acceleration 9, deceleration 9). This wash and ddH$_2$O incubation step was repeated before myelin was again fractionated in 0.32 M:0.85 M sucrose gradient and washed as before. Finally, purified myelin was resuspended in 1 mL sterile PBS and stored at –80°C until further use. Myelin was tagged with pHrodo Red according to the manufacturer's instructions. Briefly, myelin was centrifuged (10,000 × $g$ 10 min) and resuspended in 100 µL pHrodo Red succinimidyl ester (100 µg/mL in PBS 1% DMSO). Myelin was incubated for 45 min at RT in the dark before washing × 2 with 1 mL PBS. Finally, tagged myelin was resuspended in 100 µL, aliquoted, and stored at –20°C for further use.

## Microglial phagocytosis/degradation assays

Primary adult mouse microglia were isolated as described above and plated out in 96-well plates at 50,000 cells/well. Cells were cultured for 7 days or 3 days (in the case of cells from $App^{NL-G-F}$ brains) before stimulation. For phagocytosis assays, microglia were first imaged using an IncuCyte S3 Live-Cell Analysis System (Sartorius) in 'phase' and 'red' channels to gain baseline information on confluence and background fluorescence. Next, cells were stimulated with Aβ$_{1-42}$HiLyte647 (ThermoFisher, 0.5 µM), pHrodo Red $S.$ $aureus$ Bioparticles (ThermoFisher, 100 µg/mL), purified mouse myelin tagged with pHrodo Red succinimidyl ester (1:100, ThermoFisher), or isolated human synaptoneurosomes from AD brains (*Tai et al., 2014*; *Tzioras et al., 2023a*) tagged with pHrodo Red succinimidyl ester (1:20, ThermoFisher) by media change. Immediately after stimulation, cells were returned to the IncuCyte S3 for imaging in the 'phase' and 'red' channel with 9 images/well every 15 min for 2–3 hr. Images were analysed using the IncuCyte 2019B Rev2 software (Sartorius) as total integrated intensity in 'red' channel normalised to confluence in 'phase' channel.

## Human tissue

Use of human tissue for synaptoneurosome experiments above was reviewed and approved by the Edinburgh Brain Bank ethics committee and the ACCORD medical research ethics committee, AMREC (approval number 15-HV-016; ACCORD is the Academic and Clinical Central Office for Research and Development, a joint office of the University of Edinburgh and NHS Lothian). The Edinburgh Brain Bank is a Medical Research Council funded facility with research ethics committee (REC) approval (11/ES/0022).

## Cathepsin probe assays

Cathepsin probe assays were carried out as described in *Hamilton et al., 2008*. Briefly, microglia were isolated and cultured for 3 days as above before cells were lysed in RIPA buffer (60 µL/well, Merck). Next, 25 µL lysate was combined with 175 µL cathepsin probe L (Z-Phe-Arg-AMC, 45 µM, Bachem) or C (H-Gly-Phe-AMC, 56 µM, Bachem) in an assay buffer comprising 150 mM NaCl, 2 mM EDTA, 5 mM

**Table 3.** Antibodies used for immunohistochemistry.

| Antigen | Type | Species (raised) | Supplier | Cat number | Stock concentration (mg/mL) | Dilution |
|---|---|---|---|---|---|---|
| IBA1 | Primary | Rabbit | Wako | 019-19741 | 0.5 | 1:2000 |
| Aβ (6E10) | Primary | Mouse | BioLegend | 803001 | 1 | 1:1000 |
| LAMP2 | Primary | Rat | BioLegend | 108501 | 0.5 | 1:200 |
| Synaptophysin (Sy38) | Primary | Mouse | Abcam | ab8049 | Lot:1043113-1 | 1:200 |
| Anti-rabbit IgG Alexa Fluor 488 | Secondary | Goat | ThermoFisher | A-11008 | 2 | 1:500 |
| Anti-mouse IgG Alexa Fluor 555 | Secondary | Donkey | ThermoFisher | A-31570 | 2 | 1:500 |
| Anti-rat IgG Alexa Fluor 647 | Secondary | Goat | ThermoFisher | A-21247 | 2 | 1:500 |
| Anti-mouse Biotinylated | Secondary | Goat | Vector Labs | ba-9200 | 1.5 | 1:100 |

DTT, and 100 mM trisodium citrate at pH 5.5 in a black-walled, black-bottom 96-well plate (Corning). Lysates (or lysis buffer alone as a background control) were incubated for 2 hr at 37°C before reading on a fluorescence plate reader (BMG LABTECH) at excitation 360 nm, emission 460 nm with gain adjusted as necessary. Data are presented as background-corrected as relative fluorescence units.

## Immunohistochemistry

Brains from 12-month-old male and female $App^{Wt/Wt}Cst7^{+/+}$, $App^{NL-G-F}Cst7^{+/+}$, and $App^{NL-G-F}Cst7^{-/-}$ mice were isolated by terminally anaesthetising with 3% isoflurane (33.3% $O_2$ and 66.6% $N_2O$) and transcardial perfusion with ice-cold DEPC-treated 0.9% NaCl, 0.4% trisodium citrate. Brains were immediately separated down the midline then the left hemisphere placed into ice-cold 10% neutral buffered formalin (ThermoFisher). Hemi-brains were post-fixed for 48 hr at 4°C before processing for paraffin embedding. For fluorescence immunostaining used to investigate microglial and lysosomal burden, 6 µm sagittal sections were deparaffinised with 2×10 min xylene before rehydration in subsequent changes of 100%, 90%, and 70% ethanol (all 5 min). Sections were rinsed in dH₂O before antigen retrieval in 10 mM TrisEDTA pH 9 (30 min at 95°C). Next, sections were rinsed and incubated with primary antibody (*Table 3*) in PBS, 1% BSA (Merck), 0.3% Triton-X (Merck) at 4°C overnight. Sections were washed in PBS, 0.1% Tween-20 (Merck) 3×5 min before incubation with secondary antibodies (*Table 3*) in PBS, 0.1% Tween-20 (Merck), 1% BSA (Merck) at RT for 1 hr. Sections were washed again, incubated in TrueBlack Lipofuscin Autofluorescence Quencher (1:300 in 70% ethanol, Biotium) for 1 min, washed again in PBS, and rinsed in dH₂O before mounting with Fluorescence Mounting Medium (Dako). For synaptophysin staining, Thioflavin S (Merck) was added 1:200 for 5 min before mounting. Slides were imaged using an AxioImager D2 microscope (Zeiss) with a 20× objective. For quantification, two random fields of view were taken per mouse for both cortex and hippocampus. One image was taken for subiculum. One mouse was excluded from analysis as the section taken did not contain hippocampus. To remove the influence of plaque burden on the results, images were taken of approximately equal 6E10 burden. Images were analysed using a threshold analysis with QuPath v0.3.0 (*Bankhead et al., 2017*) and co-staining was quantified using Definiens Developer. For imaging and quantification of synaptophysin staining, five images were taken of a single plaque/image in $App^{NL-G-F}$ brains using a 40× oil-immersion objective with five random images in plaque-free wild-type brains. Synaptophysin coverage was calculated using ImageJ auto-threshold (default settings) and intensity within *vs.* outwith plaque area calculated using QuPath v0.3.0 (*Bankhead et al., 2017*).

For chromogenic 3,3'-diaminobenzidine (DAB) staining used to quantify amyloid burden, 6 µm sagittal sections were deparaffinised and rehydrated as above. Following antigen retrieval, endogenous peroxidase activity was blocked with 0.3% H₂O₂ for 10 min and sections were blocked with 5% normal goat serum (Vector Labs) in PBS for 1 hr at RT. 6E10 primary antibody was added at 1:500 and incubated overnight at 4°C. The next day, sections were washed and stained with biotinylated anti-mouse IgG (*Table 3*) in PBST 1% BSA for 1 hr at RT before washing and addition of ABC Elite amplification (Vector Labs) for 30 min at RT according to the manufacturer's instructions. After a further wash, slides were immersed in DAB solution (0.5 mg/mL DAB, 0.015% H₂O₂ in PBS) until stain had developed. Sections were counterstained with acidified Harris Hematoxylin (Epredia), dehydrated through increasing concentrations of ethanol and xylene, and mounted with Pertex mountant (CellPath). For imaging, slides were scanned with an Axioscanner Slide Scanner (Zeiss) in brightfield at 20× magnification. Images were analysed using a threshold analysis with QuPath v0.3.0 (*Bankhead et al., 2017*). DAB slides were stained in two batches with equal representation of groups between batches, data are presented as batch-normalised 6E10 % positive area.

## *In situ* hybridisation

*In situ* hybridisation was performed using RNAscope 2.5 HD Duplex Assay using slight modifications to the manufacturer's protocol. Freshly cut (6 µm) brain tissue was baked onto slides at 60°C for 30 min before deparaffinisation (2× xylene, 5 min) and dehydration (2×100% EtOH, 1 min). Hydrogen peroxide was applied for 10 min at RT and antigen retrieval was performed for 30 min in a pre-heated (20 min) plastic Coplin jar in a 97.5°C waterbath. Sections were left at RT overnight before placing into humidity chambers and protease plus was applied for 15 min at 37°C. Probes targeting *Cst7* (498711-C2) and *Lilrb4a* (1260291-C2) were incubated for 2 hr at 40°C, amplified according to the manufacturer's instructions, and red signal detected using RNAScope Fast Red. At this stage, antibodies raised

against IBA1 and Aβ (6E10) were added as described above and the experiment continued as with fluorescence immunohistochemistry as above. RNAScope Fast Red is detectable both in brightfield and fluorescence in 555 channel. For imaging, slides were either scanned with an Axioscanner Slide Scanner (Zeiss) in brightfield at ×20 magnification or imaged in fluorescence using an AxioImager D2 microscope (Zeiss) with a ×20 objective. For quantification, two random fields of view were taken per mouse cortex. Images were analysed for *Cst7/Lilrb4a* coverage within/outwith plaque region using QuPath v0.3.0 (*Bankhead et al., 2017*) to draw around 6E10+ area and co-localisation with IBA1 was analysed using MultipleColourAnalysis ImageJ plugin.

## Multiplex ELISA

Mouse cytokines from microglial supernatants were measured using MILLIPLEX multiplex assays as described previously (*McCulloch et al., 2022*). Briefly, MILLIPLEX MAP Mouse Cytokine/Chemokine Magnetic Bead Panel (MCYTOMAG-70K, Merck) was used to measure GM-CSF, IFN-γ, IL-1α, IL-1β, IL-2, IL-4, IL-5, IL-6, IL-12(p40), IL-33, and TNF-α. In all assays, samples were assayed as single replicates and all samples, standards, and quality controls were prepared in accordance with the manufacturer's instructions. Samples were incubated with beads on a plate for 1 hr (isotyping assay) or overnight at 4°C and washes carried out using a magnetic plate washer. Plates were analysed using a Magpix Luminex machine and Luminex xPonent software version 4.2, with a sample volume of 50 µL per well and a minimum of 50 events counted per sample.

## Randomisation and blinding

Experimenters were blinded to genotype groups throughout the study. Animals were given an experimental identifier and all samples were analysed using this coded identifier. Mice were randomly assigned to cull groups using random number generator on Microsoft Excel (which were performed in batches due to throughput for FACS) with stratification for experimental group. Order of MeX04 injection and subsequent mouse termination and tissue collection was randomised by random number generator using Microsoft Excel. Data was unblinded for analysis after experimental work was complete. There were no specific inclusion/exclusion criteria. However, mice were excluded from IHC analysis in *Figures 3 and 6* due to attrition (tissue sections were not obtainable of the specific regions required). Power calculations (α=0.05, power = 0.8) based on the primary outcome of 50% effect size on *App^NL-G-F* vs. *App^NL-G-F Cst7^-/-* %MeX04+ microglia demonstrated n=12 would be sufficient for our study.

## Statistical analyses

Data are presented as mean values + standard error of the mean (SEM). Levels of significance were $p<0.05$ (*), $p<0.01$ (**), $p<0.001$ (***). In all figures, replicates (defined by n) were biological replicates. Statistical analyses were carried out using GraphPad Prism (version 9). For RNASeq data, statistical significance was calculated using DESeq2 R package (*Love et al., 2014*). Immunohistochemistry, qPCR, cathepsin activity, and cytokine secretion were analysed with a two-way ANOVA followed by Tukey's or Sidak's post hoc comparisons, unpaired Student's t-test, or mixed effects modelling. Live-imaging data were analysed by area under the curve followed by Student's t-test or one-way ANOVA with Dunnett's multiple comparisons test. Transformations were applied where necessary. Graphs and figures were created with GraphPad Prism (version 9), VolcaNoseR (*Goedhart and Luijsterburg, 2020*), DeepVenn (*Hulsen, 2022*), and BioRender.com.

## Acknowledgements

This work is supported by the UK Dementia Research Institute which receives its funding from UK DRI Ltd, funded by the UK Medical Research Council, Alzheimer's Society and Alzheimer's Research UK and the Leducq Foundation Transatlantic Network of Excellence, Stroke IMPaCT (19CVD01). For the purpose of open access, the author has applied a CC-BY public copyright license to any Author Accepted Manuscript version arising from this submission. We would like to thank the QMRI Flow Cytometry & Cell Sorting Facility at The University of Edinburgh for assistance with FACS studies, the Shared University Research Facilities (SuRF) Histology facility at The University of Edinburgh for assistance with processing, sectioning, and slide-scanning brains, Dr Daniel Soong at The MRC Centre

for Reproductive Health, University of Edinburgh for his assistance with image analysis, and Prof. Siddharthan Chandran at The University of Edinburgh for providing the SH-SY5Y cell line.

## Additional information

### Funding

| Funder | Grant reference number | Author |
| --- | --- | --- |
| UK Dementia Research Institute | | Tara L Spires-Jones |
| Leducq Foundation | Stroke IMPaCT - 19CVD01 | Barry W McColl |

The funders had no role in study design, data collection and interpretation, or the decision to submit the work for publication.

### Author contributions

Michael JD Daniels, Conceptualization, Data curation, Formal analysis, Supervision, Investigation, Visualization, Methodology, Writing – original draft, Writing – review and editing; Lucas Lefevre, Investigation, Methodology, Project administration, Writing – review and editing; Stefan Szymkowiak, Resources, Investigation, Methodology, Writing – review and editing; Alice Drake, Laura McCulloch, Investigation, Methodology, Writing – review and editing; Makis Tzioras, Hiroki Sasaguri, Takashi Saito, Resources, Methodology, Writing – review and editing; Jack Barrington, Data curation, Formal analysis, Visualization, Methodology, Writing – review and editing; Owen R Dando, Formal analysis, Supervision, Methodology, Writing – review and editing; Xin He, Formal analysis, Methodology; Mehreen Mohammad, Methodology, Project administration; Takaomi C Saido, Resources, Methodology; Tara L Spires-Jones, Resources, Supervision, Funding acquisition, Methodology, Writing – review and editing; Barry W McColl, Conceptualization, Supervision, Funding acquisition, Methodology, Writing – original draft, Project administration, Writing – review and editing, Investigation, Resources

### Author ORCIDs

Michael JD Daniels ⓘ https://orcid.org/0000-0001-7489-5626
Laura McCulloch ⓘ http://orcid.org/0000-0002-1396-6643
Makis Tzioras ⓘ http://orcid.org/0000-0002-2660-5943
Owen R Dando ⓘ https://orcid.org/0000-0002-6269-6408
Xin He ⓘ http://orcid.org/0000-0003-1753-2591
Tara L Spires-Jones ⓘ https://orcid.org/0000-0003-2530-0598
Barry W McColl ⓘ https://orcid.org/0000-0002-0521-9656

### Ethics

Human subjects: Use of human tissue for synaptoneurosome experiments above was reviewed and approved by the Edinburgh Brain Bank ethics committee and the ACCORD medical research ethics committee, AMREC (approval number 15-HV-016; ACCORD is the Academic and Clinical Central Office for Research and Development, a joint office of the University of Edinburgh and NHS Lothian). The Edinburgh Brain Bank is a Medical Research Council funded facility with research ethics committee (REC) approval (11/ES/0022).

All animal experiments were carried out in accordance with the United Kingdom Animals (Scientific Procedures) Act 1986 and approved by the United Kingdom Home Office and the local Animal Ethical Review Group, University of Edinburgh.

### Decision letter and Author response

Decision letter https://doi.org/10.7554/eLife.85279.sa1
Author response https://doi.org/10.7554/eLife.85279.sa2

# Additional files

## Supplementary files
• MDAR checklist

## Data availability

Sequencing data obtained and reanalysed in *Figure 1* was obtained from published manuscripts. All data have been deposited in GEO: GSE89482 (*Srinivasan et al., 2020*), GSE65067 (*Wang et al., 2015*), GSE74615 (*Orre et al., 2014*), GSE117646 (*Kang et al., 2018*), GSE203202 (*Guillot-Sestier et al., 2021*), GSE127893 (*Sala Frigerio et al., 2019*). Newly generated data that support the findings of this study are uploaded to ArrayExpress (Accession ID:E-MTAB-13360).

The following dataset was generated:

| Author(s) | Year | Dataset title | Dataset URL | Database and Identifier |
|---|---|---|---|---|
| Daniels MJ, Lefevre L, Szymkowiak S, Drak A, McCulloch L, Tzioras M, Barrington J, Dando OR, He X, Mohammad M, Sasaguri H, Saito T, Saido TC, Spires-Jones TL, McColl BW | 2023 | RNA-seq to investigate sex-dependent changes in microglia in Cst7 knockout mice crossed with a mouse model of amyloid-driven Alzheimer's Disease | https://www.ebi.ac.uk/biostudies/arrayexpress/studies/E-MTAB-13360 | ArrayExpress, E-MTAB-13360 |

The following previously published datasets were used:

| Author(s) | Year | Dataset title | Dataset URL | Database and Identifier |
|---|---|---|---|---|
| Wang Y, Colonna M | 2015 | Expression data from WT and TREM2 deficient microglia in a mouse model of Alzheimer's disease | https://www.ncbi.nlm.nih.gov/geo/query/acc.cgi?acc=GSE65067 | NCBI Gene Expression Omnibus, GSE65067 |
| Friedman B, Srinivasan K, Hansen D | 2017 | Gene Expression in Cx3cr1-GFP+ Cells in PS2APP Alzheimer's Disease mice | https://www.ncbi.nlm.nih.gov/geo/query/acc.cgi?acc=GSE89482 | NCBI Gene Expression Omnibus, GSE89482 |
| Orre M, Kamphuis W, Bossers K, Hol EM | 2015 | Acutely isolated murine cortical astrocytes and microglia: Alzheimer's disease vs wildtype | https://www.ncbi.nlm.nih.gov/geo/query/acc.cgi?acc=GSE74615 | NCBI Gene Expression Omnibus, GSE74615 |
| Fryer J, Kang S | 2018 | Microglial translational profiling reveals a convergent APOE pathway from aging, amyloid, and tau | https://www.ncbi.nlm.nih.gov/geo/query/acc.cgi?acc=GSE117646 | NCBI Gene Expression Omnibus, GSE117646 |
| Mela V, Lynch MA | 2022 | Sex-related differences in Alzheimer's disease | https://www.ncbi.nlm.nih.gov/geo/query/acc.cgi?acc=GSE203202 | NCBI Gene Expression Omnibus, GSE203202 |
| Sala Frigerio C | 2019 | The major risk factors for Alzheimer's disease: Age, Sex and Genes, modulate the microglia response to Aβ plaques | https://www.ncbi.nlm.nih.gov/geo/query/acc.cgi?acc=GSE127893 | NCBI Gene Expression Omnibus, GSE127893 |

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
