## [Editor Report]

This study presents a valuable finding on the function of the gene Cst7 in sex-divergent pathological changes in microglia in a mouse model of amyloid-driven Alzheimer's disease. The evidence supporting the claims of the authors is solid, although the study would be further strengthened by validation of some of the key differentially expressed genes identified in RNA-sequencing experiments. Overall, this study offers new insight into the functional role of CST7 that is upregulated in a subset of disease-associated microglia in AD models and the human brain that will be of interest to neuroimmunologists and neuroscientists working on microglia in health and disease.

---

## [Decision Letter]

**Decision letter after peer review:**

Thank you for submitting your article "Cystatin F (Cst7) drives sex-dependent changes in microglia in an amyloid-driven model of Alzheimer's Disease" for consideration by *eLife*. Your article has been reviewed by 3 peer reviewers, one of whom is a member of our board of Reviewing Editors, and the evaluation has been overseen by and Carla Rothlin as the Senior Editor. The following individual involved in review of your submission has agreed to reveal their identity: Lucas M Cheadle (Reviewer #3).

Essential revisions:

1. The authors should validate several of the main gene expression /DE changes they report by RNA-sequencing, using an approach such as smFISH. Validation of RNA-sequencing experiments are important for confirming the identified changes in intact tissue and would allow the authors to identify which cells are expressing which genes.

Also, it is not clear if the authors performed the tissue dissociation in such a way as to avoid the aberrant activation of cells in the tissue. If not, then it is not possible to distinguish which gene expression changes occurred in vivo versus following dissection.

2. Data that supports the selectivity of Cst7 expression in microglia in AD is limited. Please address this claim by showing data from analysis of available datasets including other cell types. Given the authors use a global KO, it is important to demonstrate that Cst7-/- is in fact only affecting microglia in these experiments and comment on limitations and alternative interpretations of data in the text.

3. Addition of an earlier time point would allow the authors to assess whether Cst7 may play a role prior to late-stage disease states. For example, if the authors repeated a subset of their experiments at a 3- or 6-month time point, it would help determine whether Cst7 drives disease progression or is simply upregulated as a consequence.

4. If the central argument is that CST7 in females decreases phagocytosis and in males increases microglia activation, are there changes in amyloid plaque burden or structure in the APPNL-G-F /CST 7 KO mice compared to APPNL-G-F/CST7 WT that reflect these changes? Please address. If not, how does this affect the functional interpretation of differential expression observed in phagocytic/reactive microglia genes? Please address.

5. It is important to include a separate analysis of WT vs Cst7-/- microglia to interpret findings and address whether the effects observed in APPKI/Cst7-/- are disease-dependent effects of Cst7.

6. Data in 1B needs quantification.

7. There appears to be an error in Figure 2-S3A, the APPNL-G-F/CST7 WT mice are labeled as red, should they not be blue?

*Reviewer #1 (Recommendations for the authors):*

– Please include more discussion of the role of CST7 in AD in general and its differential role in males vs females for context and background.

– There appears to be an error in Figure2-S3A, the APPNL-G-F/CST7 WT mice are labeled as red, should they not be blue?

*Reviewer #2 (Recommendations for the authors):*

– Figure 1: Data from Guillot-Sestier vs Sala-Frigerio shows different effect of sex. I wondered if it would be worth sub-setting DAMs from Sala-Frigerio, and look specifically for sex-dependent differences in that population. It would add value to expand that analysis to other genes, ideally the DAM signature, to demonstrate whether the sexual dimorphism is specific for Cst7 or rather relates to a wider effect at cell phenotype level. Data in 1B needs quantification as otherwise is just an observation.

– The selectivity of Cst7 expression in microglia in AD is based on a single reference, and I would recommend expanding this claim by showing data from analysis of available datasets including other cell types. It is important, in my opinion, to demonstrate that Cst7-/- is in fact only affecting microglia.

– Lack of inclusion of WT Cst7-/- mice is problematic. Statistically, the study is unbalanced and any effects observed in APPKI/Cst7-/- cannot be categorically interpreted as disease-dependent effects of Cst7. I would strongly recommend including a separate transcriptomic analysis of WT vs Cst7-/- microglia.

– As designed, it is very difficult to disentangle the direct sex-specific effects from the age-related effects driven by sex, when it comes to Cst7-regulated genes or processes. We know female microglia age more rapidly into APP pathology, so the Cst7 effects reported here could be connected to the specific age of female/male microglia. My suggestion would have been to explore a different age, additional to 12 months, in order to truly ascertain whether Cst7 is sexually dimorphic or just part of a wider change, related to age, that affects females differentially.

– My comment re behaviour, from the previous section, is largely aspirational. My only hope is that the authors did perform some studies but preferred not to include those here, and that I can now plea for them to be reported. The disconnection between amyloid pathology and cognitive performance in APP models in sometimes astonishing, and it is a problem of the field not being able to see the whole picture. Ultimately, the data reported in this article would not truly be able to answer the relatively simple question of Is the disease better or worse after Cst7-/-?

– A final suggestion would be for the authors to integrate their thinking about how, molecularly, Cst7 is affecting the reported mechanisms in microglia. The reported results are largely descriptive and miss a bit the trick of describing a mechanism. The discussion covers this, to some extent, discussing potential roles of cathepsins. I understand it's often tricky to have a complete view of a mechanism, but having some ideas would help the reader.

*Reviewer #3 (Recommendations for the authors):*

One recommendation that could increase the quality and mechanistic insights in the study would be to ensure that RNA-sequencing data are acquired following protocols developed to limit the aberrant activation of microglia through tissue dissociation (see Marsh et al., Nature Neuroscience, 2022). In combination, the authors should validate several of the gene expression changes they report by RNA-sequencing using an approach such as FISH. Cell-type-resolution will be important to validate whether these changes are real, so approaches with spatial resolution are better than assays such as qPCR.

It would also be helpful for the authors to plot statistical differences for a given parameter between sexes. Even if these changes are insignificant, it would be helpful to show that on the graphs. I would also suggest that, if these changes are not significant, the authors provide a discussion of why that might be and how that might impact their conclusions.

Finally, it would be very helpful to add to the study an earlier time point that could allow the authors to assess whether Cst7 may play a role in the disease prior to late-stage disease states. For example, if the authors repeated a subset of their experiments at a 3- or 6-month time point, it would contribute substantially to their ability to determine whether Cst7 drives disease progression or is simply upregulated as a result.

---

## [Author Response]

Essential revisions:1. The authors should validate several of the main gene expression /DE changes they report by RNA-sequencing, using an approach such as smFISH. Validation of RNA-sequencing experiments are important for confirming the identified changes in intact tissue and would allow the authors to identify which cells are expressing which genes.

We focussed attention on *Lilrb4* given its inclusion in Figure 2E showing sex-dependent patterns altered in disease and its relevance to myeloid cell activation. Using combined single-molecule detection FISH (RNAscope) and immunofluorescence we observed that *Lilrb4* is expressed in plaque-associated IBA1+ cells (Figure 2. – figure supplement 3). Quantification of IBA1+*Lilrb4*+ showed the equivalent pattern to our microglial RNAseq data thus confirming findings *in situ*. We also attempted a similar approach with probes for *Il1b*, another gene we showed has sex-dependent patterns in the disease condition, however we could not get probe labelling to work, including in positive control samples.

Also, it is not clear if the authors performed the tissue dissociation in such a way as to avoid the aberrant activation of cells in the tissue. If not, then it is not possible to distinguish which gene expression changes occurred in vivo versus following dissection.

We took standard precautions to minimise the risk of aberrant ex vivo cell activation, including maintaining cells on ice during non-enzyme steps of the procedure and carrying out preps in small batches to minimise time taken from removal of brain to purification of microglial RNA. Importantly, we also validated key expression data by *in situ* methods as above (Figure 2. – figure supplement 3) thus eliminating dissection-induced effects. Additionally, when performing qPCR on microglia from non-disease mice to test the disease-specific role of *Cst7*-dependent gene regulation we did not observe the same gene changes (Figure 2. – figure supplement 4) which, if such changes were dependent on tissue dissociation, we would expect to observe in naive or disease animals. We utilised the resources provided by Marsh et al. 2022 to search for overlap between enzyme-induced genes and our DEG lists from our key comparisons. We found the enzyme-induced gene set had very minimal overlap with any of our comparisons with overlap of only 4 genes between enzyme-induced genes and *Cst7*-dependent genes in males and no overlap between enzyme-induced genes and *Cst7*-dependent genes in females (see Author response image 1). We would further point out that the disease-induced microglial RNAseq profile in the *App^NL-G-F^ Cst7^+/+^* (i.e. disease WT) condition mirrors those observed previously by multiple methods including in situ profiling (Zeng et al. 2023 – PMID: 36732642) and RiboTag approaches (Kang et al. 2018 – PMID: 30082275). With a combination of these approaches, we believe it extremely unlikely that the gene changes we observe our due to artifacts induced by dissection.

**Author response image 1. sa2fig1:** 

2. Data that supports the selectivity of Cst7 expression in microglia in AD is limited. Please address this claim by showing data from analysis of available datasets including other cell types. Given the authors use a global KO, it is important to demonstrate that Cst7-/- is in fact only affecting microglia in these experiments and comment on limitations and alternative interpretations of data in the text.

We agree this is an important point and now provide multiple sources of evidence that *Cst7* is almost exclusively derived from microglia (and likely only in disease conditions). *Cst7* is expressed only in the microglia/macrophage sample from bulk RNAseq on sorted CNS cell types (Zhang et al., 2014 – PMID: 25186741/BrainRNASeq) and in scRNAseq brain cell atlases (Saunders et al., 2018 – PMID: 30096299/DropViz). All these sources indicate expression is negligible in steady-state as we observe in the current study, including by RNAseq (Figure 2. – figure supplement 3) and *Cst7* FISH (Figure 1. – figure supplement 1). Therefore, relative cell expression of *Cst7* in steady-state must be interpreted with this in mind but nonetheless if it is expressed it appears restricted to brain macrophages and further to microglia (e.g. no co-expression with border macrophage marker gene expressing cells in Saunders et al., 2018 – PMID: 30096299/DropViz). Our FISH data show *Cst7* exclusively expressed in the *App^NL-G-F^* condition and localised only to plaque-associated IBA1+ cells (without perivascular location) (Figure 1B-E and Figure 1. – figure supplement 1). An independent scRNAseq dataset of CNS macrophage populations also shows above-background expression in reactive (DAM) microglia only (Van Hove et al., 2019 – PMID: 31061494) and a recent spatial transcriptomics analysis of TauPS2APP mice shops the high *Cst7* expression restricted to peri-plaque microglia (Zeng et al., 2023 – PMID: 36732642) (Figure 1. – figure supplement 1). We also highlight that *Cst7* is expressed in reactive microglia but not monocyte-derived macrophages in brain pathology where these cells co-locate (Patir et al., 2023, BioRXiv, Figure 8). We have updated figures with the additional data from independent datasets and further comment on the weight of evidence indicating effects of *Cst7* deletion are likely explained by a microglial mechanism.

3. Addition of an earlier time point would allow the authors to assess whether Cst7 may play a role prior to late-stage disease states. For example, if the authors repeated a subset of their experiments at a 3- or 6-month time point, it would help determine whether Cst7 drives disease progression or is simply upregulated as a consequence.

We believe this point relates to a comment from reviewer 3 to whom we also reply below. Our data strongly support a mechanism whereby early disease drives *Cst7* expression which then exerts disease-influencing effects on further progression as shown by the effects we see of *Cst7* deletion on microglial phenotype and disease pathology. We expect this is reflective more broadly of how microglia may be involved in AD-like progression i.e. early trigger(s) provoke microglial reactivity and thereafter modulators of this microglial reaction then influence subsequent disease progression. We accept that exploring additional timepoints both before and after the 12-month point we selected could potentially provide additional insight; unfortunately, we do not have the resources available to conduct additional timepoint studies. Nonetheless, we believe the existing conclusions would not likely be materially altered. *Cst7* expression is low until 6-12 months (see data from Sala-Frigerio et al.) making a 12 month timepoint a rational age to evaluate both the impact on cell phenotype/functions and disease pathology. At this age, cells inducing *Cst7* from a negligible baseline (see comments above) have had the temporal window to affect disease pathology, as we find in the current study in a sex-dependent manner. Overall, we believe the fact that genetic deletion of a single (microglia-specific) gene has an effect on microglial phenotype and disease pathology is strong evidence that the effects observed are due to the gene itself and not simply a result of upregulation as a consequence.

4. If the central argument is that CST7 in females decreases phagocytosis and in males increases microglia activation, are there changes in amyloid plaque burden or structure in the APPNL-G-F /CST 7 KO mice compared to APPNL-G-F/CST7 WT that reflect these changes? Please address. If not, how does this affect the functional interpretation of differential expression observed in phagocytic/reactive microglia genes? Please address.

This comment relates to a query from Reviewer #1 – we emphasise the data already presented in Figure 6 and Figure 6 – figure supplement 2 showing altered Aβ burden (6E10 staining) and plaque count (MeX04) but no change in plaque area. Regarding the functional interpretation of *Cst7*-dependent gene changes in microglia beyond the endolysosomal function we present in figures 3-5, we have included additional data using simple immunohistochemistry, as suggested by the reviewer, to assess synapse abundance. We show loss of Sy38 coverage around plaques (Figure 6I) and a moderate but significant decrease in coverage between *App^NL-G-F^/Cst7^-/-^* vs *App^NL-G-F^* brains only in females (Figure 6J). This reflects the effect observed with plaque coverage whereby we observe increased burden in *App^NL-G-F^/Cst7^-/-^* vs *App^NL-G-F^* females but not males (Figure 6B-F) suggesting the increased plaque burden in *Cst7^-/-^* female mice may lead to increased synapse loss. We would also emphasise that altered expression of phagolysosomal genes could affect disease in ways beyond interactions with amyloid and synapses.

5. It is important to include a separate analysis of WT vs Cst7-/- microglia to interpret findings and address whether the effects observed in APPKI/Cst7-/- are disease-dependent effects of Cst7.

We agree this is an important element to the interpretation of the manuscript. We have now compared expression of genes for which we found sex/disease-dependent differential expression of *Cst7* in our original analysis (e.g. *Il1b, Lilrb4a*) in non-disease microglia isolated from WT and *Cst7*-/- mice and this shows no significant effect of *Cst7* deletion on either gene (Figure 2. – figure supplement 3). Moreover, the disease-specific expression pattern described above and in Figure 1. – figure supplement 1 and our own data (Figure 2 – figure supplement 3) with *Cst7* minimally expressed at baseline and upregulated in disease supports that *Cst7* will mediate effects in disease but not baseline state. Consistent with this, we saw no effect of *Cst7* deficiency in our in multiple vitro assays using microglia derived from non-disease mice (Figure 5 – figure supplement 1-3). In contrast, assays using microglia derived from 12-month-old *App^NL-G-F^* mice revealed functional effects of *Cst7* deficiency (Figure 5). Together, these findings emphasise a disease-dependent role for *Cst7*.

6. Data in 1B needs quantification.

We have provided quantification of our own newly-generated *Cst7* FISH data now presented in Figure 1. Here we show that *Cst7* is expressed specifically around the plaques and overlaps with IBA1+ cells.

7. There appears to be an error in Figure 2-S3A, the APPNL-G-F/CST7 WT mice are labeled as red, should they not be blue?

Thank you for picking this up – we have corrected the figure.

Reviewer #1 (Recommendations for the authors):– Please include more discussion of the role of CST7 in AD in general and its differential role in males vs females for context and background.

We believe we have covered this topic in the introduction (third paragraph). However, we have added recent references to the limited studies on the functional role of *Cst7* in AD and to the regulation of *Cst7* in ageing.

– There appears to be an error in Figure2-S3A, the APPNL-G-F/CST7 WT mice are labeled as red, should they not be blue?

We thank the reviewer for spotting this error and have rectified the labelling mistake.

We also draw attention to further revisions conducted beyond these recommendations in our point-by-point response to all comments above from Reviewer #1.

Reviewer #2 (Recommendations for the authors):– Figure 1: Data from Guillot-Sestier vs Sala-Frigerio shows different effect of sex. I wondered if it would be worth sub-setting DAMs from Sala-Frigerio, and look specifically for sex-dependent differences in that population. It would add value to expand that analysis to other genes, ideally the DAM signature, to demonstrate whether the sexual dimorphism is specific for Cst7 or rather relates to a wider effect at cell phenotype level. Data in 1B needs quantification as otherwise is just an observation.

As requested, we have sub-setted the DAMs (ARMs in this case) from Sala-Frigerio et al. and investigated sex-specific effects (Figure 1 —figure supplement 1A). Effect sizes overall are low, suggesting the main effect observed is due to the proportion of ARMs being (marginally) greater in female than male rather than the ARMs identity being different. However, we observe a number of genes more greatly expressed in the female ARM than male including *Spp1*, *Cst7*, and *Gpnmb* (as well as sex-chromosome genes). We have added this analysis to Figure 1A.

Regarding the data in 1B please see out comment above. Also copied here:

We have provided quantification of our own newly-generated *Cst7* FISH data now presented in Figure 1. Here we show that *Cst7* is expressed specifically around the plaques and overlaps with IBA1+ cells.

– The selectivity of Cst7 expression in microglia in AD is based on a single reference, and I would recommend expanding this claim by showing data from analysis of available datasets including other cell types. It is important, in my opinion, to demonstrate that Cst7-/- is in fact only affecting microglia.

Please see our response to Reviewer #1 above, repeated here:

We agree this is an important point and now provide multiple sources of evidence that *Cst7* is almost exclusively derived from microglia (and likely only in disease conditions). *Cst7* is expressed only in the microglia/macrophage sample from bulk RNAseq on sorted CNS cell types (Zhang et al. 2014 – PMID: 25186741/BrainRNASeq) and in scRNAseq brain cell atlases (Saunders et al. 2018 – PMID: 30096299/DropViz). All these sources indicate expression is negligible in steady-state as we observe in the current study, including by RNAseq (Figure 2. – figure supplement 3), *Cst7* FISH (Figure 1. – figure supplement 1) and qPCR on isolated microglia in culture (Figure 5B). Therefore, relative cell expression of *Cst7* in steady-state must be interpreted with this in mind but nonetheless if it is expressed it appears restricted to brain macrophages and further to microglia (e.g. no co-expression with border macrophage marker gene expressing cells in Saunders et al. 2018 – PMID: 30096299/DropViz). Our FISH data show *Cst7* exclusively expressed in the *App^NL-G-F^* condition and localised only to plaque-associated IBA1+ cells (without perivascular location) (Figure 1B-E and Figure 1. – figure supplement 1). An independent scRNAseq dataset of CNS macrophage populations also shows above-background expression in reactive (DAM) microglia only (Van Hove et al. 2019 – PMID: 31061494) and a recent spatial transcriptomics analysis of TauPS2APP mice shops the high *Cst7* expression restricted to peri-plaque microglia (Zeng et al. 2023 – PMID: 36732642) (Figure 1. – figure supplement 1). We also highlight that *Cst7* is expressed in reactive microglia but not monocyte-derived macrophages in brain pathology where these cells co-locate (Patir et al. 2023, BioRXiv, Figure 8). We have updated figures with the additional data from independent datasets and further comment on the weight of evidence indicating effects of *Cst7* deletion are likely explained by a microglial mechanism.

– Lack of inclusion of WT Cst7-/- mice is problematic. Statistically, the study is unbalanced and any effects observed in APPKI/Cst7-/- cannot be categorically interpreted as disease-dependent effects of Cst7. I would strongly recommend including a separate transcriptomic analysis of WT vs Cst7-/- microglia.

We agree this is an important element to the interpretation of the manuscript. We have now compared expression of genes for which we found sex/disease-dependent differential expression of *Cst7* in our original analysis (e.g. *Il1b, Lilrb4a*, *Cst7*) in non-disease microglia isolated from WT and *Cst7*-/- mice and this shows no significant effect of *Cst7* deletion on any gene (Figure 2. – figure supplement 3). Moreover, the disease-specific expression pattern described above and in Figure 1. – figure supplement 1 (with *Cst7* minimally expressed at baseline and upregulated in disease) supports that *Cst7* will mediate effects in disease but not baseline state. Consistent with this, we saw no effect of *Cst7* deficiency in our in multiple vitro assays using microglia derived from non-disease mice (Figure 5 – figure supplement 1-3). In contrast, assays using microglia derived from 12-month old *App^NL-G-F^* mice revealed functional effects of *Cst7* deficiency (Figure 5). Together, these findings emphasise a disease-restricted role for *Cst7*.

– As designed, it is very difficult to disentangle the direct sex-specific effects from the age-related effects driven by sex, when it comes to Cst7-regulated genes or processes. We know female microglia age more rapidly into APP pathology, so the Cst7 effects reported here could be connected to the specific age of female/male microglia. My suggestion would have been to explore a different age, additional to 12 months, in order to truly ascertain whether Cst7 is sexually dimorphic or just part of a wider change, related to age, that affects females differentially.

We agree this is a challenging element to disentangle however would reiterate that whether *Cst7* expression in disease is sexually dimorphic is a separate question from whether deleting *Cst7* has a sexually dimorphic impact on microglial phenotype and other disease features – indeed, that *Cst7* itself seems to not be expressed significantly differently between male and female *App^NL-G-F^* microglia in our data , along with other data, favours that the impact of deleting *Cst7* is a direct/intrinsic sex effect and not simply a sex effect caused by sex-related stepped ageing trajectories.

Our response above to Reviewer #1 is repeated here for consistency:

This is an interesting question and while we acknowledge that empirically addressing with a later timepoint could add insight, we believe it would actually need multiple closely-spaced timepoints as choosing what single later timepoint would be optimal is difficult to judge (and likely not possible at all) for reasons below. We also believe data already published combined with our observations show it is most-likely a cell-intrinsic effect that explains our sex-specific differences.

First, we emphasize the acceleration of the microglial phenotype in female *App^NL-G-F^* mice previously published is fairly subtle and relative rather than absolute e.g. the DAM/ARM microglia state represents ~50% of all microglia in male and ~55% of all microglia in females at 12 months old therefore both sexes have similarly abundant microglia in the state that most highly express *Cst7.* Indeed, after the age at which DAM/ARM state microglia appear in appreciable numbers (~ 6 months), both females and males both have an abundance of them. It is important to note that a 12-month male is far more “progressed” than a 6-month female hence the stepped age effect is temporally short.

Second, *Cst7* deletion in the *App^NL-G-F^* mice condition caused qualitative differences affecting distinct genes and/or overlapping genes moving in different directions between female and male mice (Figure 2, Figure 2-figure supplement 3) – if a stepped age effect explained sex differences from *Cst7* deletion, given that it could only be stepped by a very short timeframe (several weeks maximum) from reasoning above, we would expect to see similar qualitative changes but of different magnitude in female and male mice arising from *Cst7* deletion; this is not the pattern we see.

Third, beyond 12 months old, regression from ARM/DAM actually occurs, again making it unlikely males would “catch up” with females to show the same profile from *Cst7* deletion but just at an older age – practically, this also complicates choosing a single later timepoint (and age-related systemic morbidity emerges as a potential confounder as well).

In summary, while the acceleration of the DAM signature in female microglia offers an intriguing possible explanation to our observation of sexual dimorphism in response to deletion of one of the key genes in this signature, we believe it more likely that intrinsic effects are responsible for the *Cst7* deletion sex-related impact. Taking the alternative perspective, even if a stepped age effect in the underlying progression of the model could explain our findings, this would need multiple timepoints with short gaps between (e.g. monthly at 12, 13, 14, 15 months old) to provide the temporal resolution to expose this pattern; we would not have the resources to conduct such a resource-intensive and lengthy study. We hope this reasoning appears logical and conscious of the importance to convey this in our manuscript we have revised the Discussion to as concisely as possible capture some key points outlined above.

– My comment re behaviour, from the previous section, is largely aspirational. My only hope is that the authors did perform some studies but preferred not to include those here, and that I can now plea for them to be reported. The disconnection between amyloid pathology and cognitive performance in APP models in sometimes astonishing, and it is a problem of the field not being able to see the whole picture. Ultimately, the data reported in this article would not truly be able to answer the relatively simple question of Is the disease better or worse after Cst7-/-?

We absolutely agree with the reviewer that a behavioural/cognitive readout would be desirable. We considered from the outset but there is no behavioural test we have found with significantly robust behavioural alteration in our hands using the *App^NL-G-F^* model. Indeed, there is no clear consensus in the literature for a robust behavioural readout with this model (Sakakibara et al. 2019 – PMID: 30894120, Figure 4; also personal communication with researchers across two international networks we are part of). However, to help provide insight to the key question of “Is the disease better or worse after *Cst7^-/-^*" we have performed immunohistochemistry as suggested by Reviewer #1 and now reported in Figure 6I and J and Figure 6 – figure supplement 1. As described above, we show loss of Sy38 coverage around plaques (Figure 6I) and a modest but significant decrease in coverage between *App^NL-G-F^/Cst7^-/-^ vs. App^NL-G-F^* brains only in females. This reflects the effect observed with plaque coverage whereby we observe increased burden in *App^NL-G-F^/Cst7^-/-^* vs *App^NL-G-F^* females but not males (Figure 6B-F) suggesting the increased plaque burden in *Cst7^-/-^* female mice may lead to increased synapse loss.

– A final suggestion would be for the authors to integrate their thinking about how, molecularly, Cst7 is affecting the reported mechanisms in microglia. The reported results are largely descriptive and miss a bit the trick of describing a mechanism. The discussion covers this, to some extent, discussing potential roles of cathepsins. I understand it's often tricky to have a complete view of a mechanism, but having some ideas would help the reader.

We believe we have addressed the molecular mechanisms by which *Cst7* deletion could lead to the effects we observe in this study in the discussion (see second paragraph). However, we have added to this to further integrate thinking on possible mechanisms involved.

Reviewer #3 (Recommendations for the authors):One recommendation that could increase the quality and mechanistic insights in the study would be to ensure that RNA-sequencing data are acquired following protocols developed to limit the aberrant activation of microglia through tissue dissociation (see Marsh et al., Nature Neuroscience, 2022). In combination, the authors should validate several of the gene expression changes they report by RNA-sequencing using an approach such as FISH. Cell-type-resolution will be important to validate whether these changes are real, so approaches with spatial resolution are better than assays such as qPCR.

Please see response above where we provide *in situ* data on *Cst7* and *Lilrb4a* in non-dissociated tissue to show microglial expression around plaques and mirrored patterns of expression in experimental groups as per our RNASeq data on sorted microglia.

It would also be helpful for the authors to plot statistical differences for a given parameter between sexes. Even if these changes are insignificant, it would be helpful to show that on the graphs. I would also suggest that, if these changes are not significant, the authors provide a discussion of why that might be and how that might impact their conclusions.

Please see response above where we comment further on the interaction effect we observe between sex and *Cst7* status. We have added further comment in the discussion on the manuscript and we have added the gene, sex and interaction effects from the main ANOVA onto the figures.

Finally, it would be very helpful to add to the study an earlier time point that could allow the authors to assess whether Cst7 may play a role in the disease prior to late-stage disease states. For example, if the authors repeated a subset of their experiments at a 3- or 6-month time point, it would contribute substantially to their ability to determine whether Cst7 drives disease progression or is simply upregulated as a result.

Please see response above where we discuss the utility of assessing *Cst7* at an earlier timepoint. We believe the fact that *Cst7* deletion affects key disease readouts such as plaque burden and synapse loss is direct evidence that *Cst7* is causally affecting disease progression.